# Pleistocene chronology and history of hominins and fauna at Denisova Cave

Zenobia Jacobs [1,2] ✉, Elena I. Zavala[3,4] ✉, Bo Li [1,2], Kieran O'Gorman [1], Michael V. Shunkov[5], Maxim B. Kozlikin [5], Anatoly P. Derevianko [5], Vladimir A. Uliyanov[6], Paul Goldberg [1,7], Alexander K. Agadjanian[8], Sergei K. Vasiliev[5], Frank Brink[9], Stéphane Peyrégne [3], Viviane Slon [3,10,11,12], Svante Pääbo [3], Janet Kelso [3], Matthias Meyer [3] ✉ & Richard G. Roberts [1,2] ✉

Denisova Cave in southern Siberia is the only site known to have been occupied by Denisovans, Neanderthals and modern humans. The cave consists of three chambers (Main, East and South), with the archaeological assemblages and remains of hominins, fauna and flora recovered from Main and East Chambers being the most thoroughly investigated to date. Here we report the results of analyses of the Palaeolithic artefacts, faunal remains and hominin and mammalian mitochondrial (mt) DNA recovered from renewed excavations in South Chamber. We construct a calendar-year time scale for the stratified Pleistocene deposits from optical dating of the sediments. The timing of hominin occupation and major turnovers in the mtDNA of Denisovans and large mammals largely accords with the patterns detected in Main and East Chambers. Time gaps in those sequences are partly filled by the South Chamber data and the sediment DNA record of Denisovans after 80,000 years ago is more than doubled in size. We combine the sediment dating and DNA records for all three chambers to reveal the whole-of-cave history of this unique site and the climatic conditions experienced by hominins and fauna over the past 300,000 years, including potential changes in habitat suitability for Denisovans and Neanderthals.

Situated in the foothills of the Altai Mountains, Denisova Cave was inhabited by two groups of archaic hominins—Denisovans and Neanderthals—during the Middle and Late Pleistocene, and by ancient modern humans from at least 45 thousand years ago (ka)[1–3]. Current understanding of their occupational history is based largely on the Middle and Upper Palaeolithic assemblages[4–7] recovered from the stratified deposits in two of the three chambers of the cave and on the skeletal remains and mitochondrial (mt) and nuclear DNA retrieved from the hominin fossils and cave sediments[2,3,8–25].

[1]Centre for Archaeological Science, School of Science, University of Wollongong, Wollongong, NSW, Australia. [2]Australian Research Council Centre of Excellence for Australian Biodiversity and Heritage, University of Wollongong, Wollongong, NSW, Australia. [3]Max Planck Institute for Evolutionary Anthropology, Leipzig, Germany. [4]Department of Molecular and Cell Biology, University of California, Berkeley, CA, USA. [5]Institute of Archaeology and Ethnography, Russian Academy of Sciences, Siberian Branch, Novosibirsk, Russia. [6]Lomonosov Moscow State University, Moscow, Russia. [7]Institut für Naturwissenschaftliche Archäologie, Universität Tübingen, Tübingen, Germany. [8]Borissiak Palaeontological Institute, Russian Academy of Sciences, Moscow, Russia. [9]Centre for Advanced Microscopy, Australian National University, Canberra, ACT, Australia. [10]Department of Anatomy and Anthropology, The Gray Faculty of Medical & Health Sciences, Tel Aviv University, Tel Aviv, Israel. [11]Department of Human Molecular Genetics and Biochemistry, The Gray Faculty of Medical & Health Sciences, Tel Aviv University, Tel Aviv, Israel. [12]The Dan David Center for Human Evolution and Biohistory Research, Tel Aviv University, Tel Aviv, Israel. ✉e-mail: zenobia@uow.edu.au; elena.zavala@sund.ku.dk; mmeyer@eva.mpg.de; rgrob@uow.edu.au

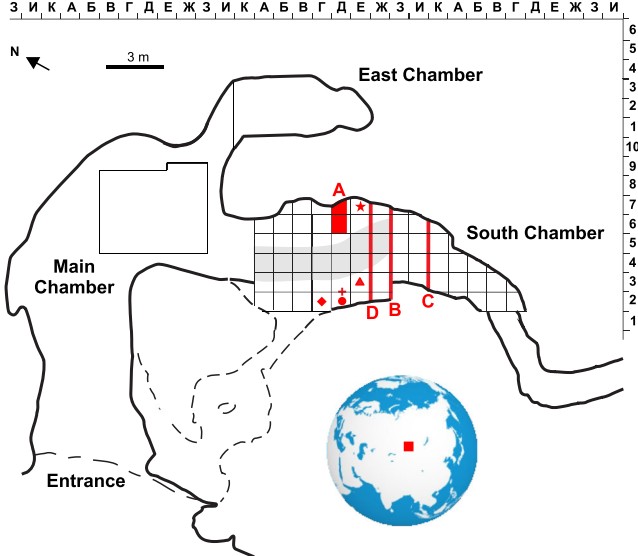

**Fig. 1 | Location map and site plan of Denisova Cave.** Plan of the cave interior, showing Main, East and South Chambers, and locations in South Chamber of Profiles A–D (red lines), Denisovan fossils (red symbols: Denisova 4, diamond; Denisova 13, cross; Denisova 22, star; Denisova 25, circle) and the deer tooth pendant and mammoth ivory figurine (red triangle). The line for Profile A is thicker than the others because this profile was cleaned back and sampled over three excavation seasons. The grey shading denotes the approximate position of the ridge of spalled rocks. Grid coordinates for excavation squares are shown along the top and right sides, and the corresponding squares (consisting of a Cyrillic letter and a number) are shown at the top of the profiles in Figs. 2, 6. Location of Denisova Cave in the Altai region of southern Siberia is shown inset (red square). Globe image created using ArcGIS Pro 3.4.3 using basemap sources Esri, TomTom, FAO, NOAA and USGS.

Excavations in Main and East Chambers have so far yielded six fossils of Denisovans, four of Neanderthals, and a bone fragment of a Neanderthal–Denisovan offspring (Denisova 11), as well as mtDNA fragments of Denisovans, Neanderthals and ancient modern humans in 160 sediment samples. Fossil remains and mtDNA sequences recovered from sediments also provide evidence that cave-dwelling animals were present at various times[1,3,22,23,26–29]. The chronology of these stratigraphic sequences has been established from optical dating of sediments (92 ages)[1] and radiocarbon ($^{14}$C) dating of bone, tooth and charcoal younger than -50 ka (60 ages)[2,13,30].

The stratigraphic sequence in South Chamber has been studied less extensively, with a provisional chronology for the Pleistocene deposits developed from six $^{14}$C ages[13,31] and 11 optical ages[1]. Excavations recommenced in 2017, revealing a ridge of spalled rock running along its central axis (Fig. 1). This has resulted in deformation and possible mixing of layers in places, especially on the left (northeast) side of the ridge and in the middle of the chamber. Parts of the uppermost Pleistocene layers have also been affected by post-depositional mineralization of phosphates derived from the overlying Holocene deposits[1,3,32,33], and possibly by earthquake-induced deformations[34,35]. The Pleistocene deposits on the right (southwest) side of the ridge show the least disturbance and clearest stratigraphy. A variety of stone and bone tools have been recovered from the Middle and Upper Palaeolithic deposits (Supplementary Section 1), including a range of personal ornaments from the latter[5,31,36–42], as well as two Denisovan teeth (Denisova 4[10,13,16] and Denisova 25[25]), two fragments of Denisovan skull (Denisova 13[2,20,24] and Denisova 22), and ancient hominin mtDNA in 15 of 202 sediment samples collected in 2017[3].

Here we report an additional 49 optical ages for sediments collected during excavations in South Chamber in 2017–2019, and mtDNA

data for a further 235 sediment samples collected in 2019 and analyzed for genetic traces of ancient hominins and mammalian fauna. We combine these records with those of the artefact assemblages[43–52] and skeletal remains of large and small mammals[53–59] to establish a timeline for human and faunal occupation of South Chamber, and present microstratigraphic (micromorphological) information[23,27,60,61] on the formation history and structural integrity of the deposits. We then generate a timeline for the stratigraphic sequences in Main, East and South Chambers, based on a total data set of 150 optical ages (53, 37 and 60, respectively), and combine this with the archaeological, palaeontological and ancient hominin and faunal mtDNA data obtained from 963 sediment samples (274, 252 and 437 from Main, East and South Chambers, respectively) to reconstruct the whole-of-cave Pleistocene history of Denisova Cave.

## Results
### Stratigraphy and micromorphology
Sediment samples for optical dating, ancient DNA and micromorphological analysis were collected from the southeast profiles of four stratigraphic sections in South Chamber, referred to here as Profiles A–D (Figs. 1, 2; Supplementary Fig. 26). Profile A is located on the left side of the ridge, whereas Profiles B–D span the full width of the chamber. Two DNA samples were also collected from clumps of sediment attached to a deer tooth pendant (data published in ref. 31) and mammoth ivory figurine[40] found in layer 11 on the right side of the ridge (same excavation square), as well as three sediment samples collected from directly beneath a fragment of hominin skull (Denisova 22, unpublished) recovered in 2019 from the left side of the ridge (Fig. 1). The latter samples were assigned to layer 13 during excavation, but we cannot exclude their possible association with layer 12, given the complex stratigraphy in this area. The sediments beneath Denisova 22 were also sampled for optical dating and micromorphological analysis. Here we use the numbering scheme currently applied to the layers in South Chamber (Supplementary Data 1). Their sedimentological characteristics and reconstructed sequence of post-depositional deformations are described in Supplementary Section 1.

The lowermost unit in Profile A (layer 19) is a yellowish-brown silty clay, originally deposited in phreatic conditions, with rip-up clasts of reworked sediment and grey (gleyed) patches indicative of water-logged conditions (Supplementary Fig. 1). Areas of light brown sediment are interpreted as animal burrows or deformation features resulting primarily from subsidence, infilled by the overlying deposits that consist of poorly sorted silty sand with variable amounts of reworked material and limestone fragments spalled from the cave walls and roof. These deposits could not be confidently assigned layer numbers based on the lithostratigraphy and are referred to as 'deformed Middle Palaeolithic' (dMP) deposits[3]. Evidence of burrowing and microscopic traces of bioturbation and syn-depositional slumping (coprolites, bone fragments, reworked grains of phreatic clay) are particularly prominent at the unconformable contact between layer 19 and the dMP deposits (Fig. 3a–d).

Profiles B and C expose the uppermost Pleistocene layers and overlying Holocene deposits. The deposits closest to the cave walls (tentatively recognized as layer 11 during excavation[1]) consist of brownish-grey, fine-grained sediments with abundant limestone fragments, separated by reddish-brown, poorly sorted silty clay (layers 12 and 9). The latter sediments (particularly in layer 9) have been extensively disturbed by burrowing and phosphate mineralization. These 'phosphate deformation deposits' (pdd-12 and −9)[3] also have microscopic features typical of freeze–thaw processes (platy structures, rounded aggregates), most likely associated with modern seasonal frost, and include fragments of bone, tooth, coprolite, limestone (some with phosphatized rims), siliceous rocks (schist, siltstone) and phosphate grains, veins and compound

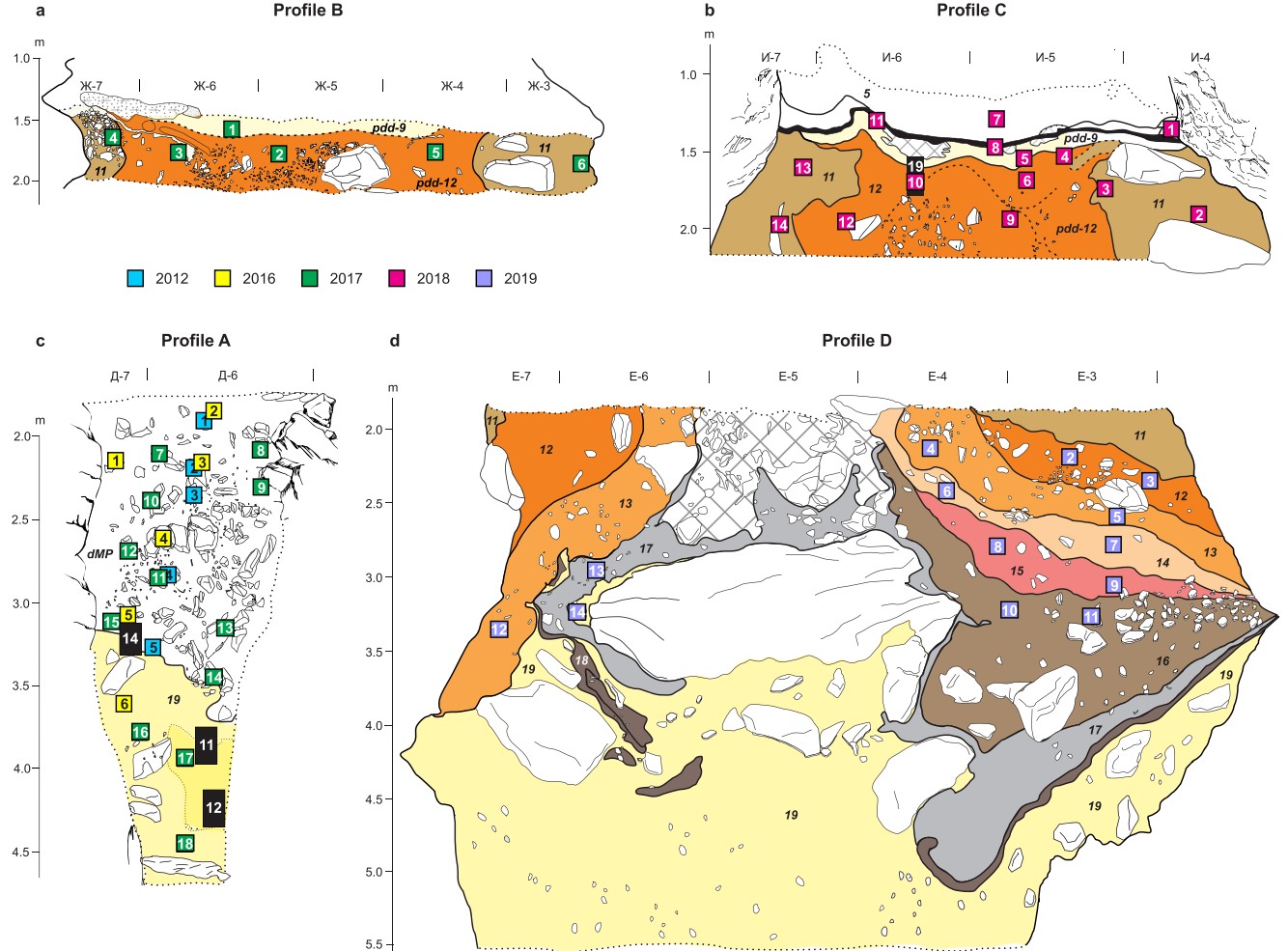

**Fig. 2 | Stratigraphy of South Chamber deposits and locations of optical dating and micromorphology samples. a–d** Stratigraphy of southeast faces of Profiles B, C, A and D, respectively. Locations of optical dating samples are shown as squares and sample numbers are enclosed within the squares, which are colour-coded by year of collection; locations of samples DCS19-1a, −1b and −1c are indicated by coloured squares and dashed lines in Supplementary Fig. 3a. Black rectangles indicate locations of micromorphology samples in Profiles A and C, with the sample codes inset in white; locations of three other micromorphology samples are shown in Fig. 3e (DEN18-18), Supplementary Fig. 1a (DEN18-13) and Supplementary Fig. 3a (DEN19-1). Stratigraphic layers are distinguished by arbitrary colours and layer numbers are displayed in italics. Rocks, Holocene layers 7 and 5, and areas where layer attribution is uncertain (dMP, deformed Middle Palaeolithic deposits in Profile A; cross-hatched area in Profile D) are shown in white; pdd, phosphate deformation deposits. Stippled lines denote sedimentary deposits that extend beyond the borders of the profiles. Samples are shown in relative stratigraphic position in Profile A, which represents a composite of the stratigraphic sequences sampled over three excavation seasons. Profile D was excavated to a depth of -3.5 m at the time of sampling. Vertical scales denote elevation (in metres) below cave datum and excavation squares are 1 m wide.

nodules (Fig. 3e–h; Supplementary Fig. 2). Subsidence and, possibly, large earthquakes[35] led to the deformation of layers 12 and 11, with materials from older layers incorporated in places. Layer 9 was formed from the erosion, mixing and redeposition of sediments from layers 12 and 11, with post-depositional phosphatization resulting in pdd-9 and, in the phosphatized parts of layer 12, pdd-12.

Profile D is located closer to the entrance than Profiles B and C and exposes layers 19–11; the layers in this profile have not been noticeably affected by phosphatization. The stratigraphy is clearest on the right side of Profile D, where layers 16–11 consist of poorly sorted, light brown and reddish-brown silty sand with abundant inclusions of spalled limestone. On the left side, remnants of layers 19–17 are wrapped around the rocks in the middle of the chamber and layers 13 and 12 dip steeply towards the cave wall; layer 18 was too thin to collect optical dating or sediment DNA samples from the exposed profile. The reddish-brown sediments adjacent to Denisova 22 (layer 13) are a poorly sorted mixture of sand, silt and clay, with inclusions of limestone, siliceous rocks, bone and rounded aggregates of sediment reworked from layer 19 (Supplementary Fig. 3).

## Chronology of sediment deposition

Optical ages are presented in Fig. 4 and Supplementary Fig. 22, and equivalent dose ($D_e$) values, environmental dose rates and other supporting data are provided in Supplementary Section 2. $D_e$ values were estimated using the optically stimulated luminescence (OSL) and post-infrared infrared stimulated luminescence (pIRIR) signals from sand-sized grains of quartz and potassium-rich feldspar (K-feldspar), respectively (see Methods). Single-grain measurements were made on 57 of 60 samples (95%), and the other three samples (all older than 300 ka) were measured using a multi-grain pIRIR procedure. For 28 samples, reliable OSL and pIRIR ages were obtained from both minerals; we combined the paired ages to determine a weighted mean age for each of these samples.

In general, the South Chamber sequence proved more challenging to date than those in Main and East Chambers, due to more extensive phosphatization and burrowing and the higher proportion of samples containing material reworked from older deposits. Nonetheless, 50 samples (83%) have $D_e$ distributions dominated by a single population of grains (after removing any statistically significant outliers) or consist of two or three discrete components, one of which

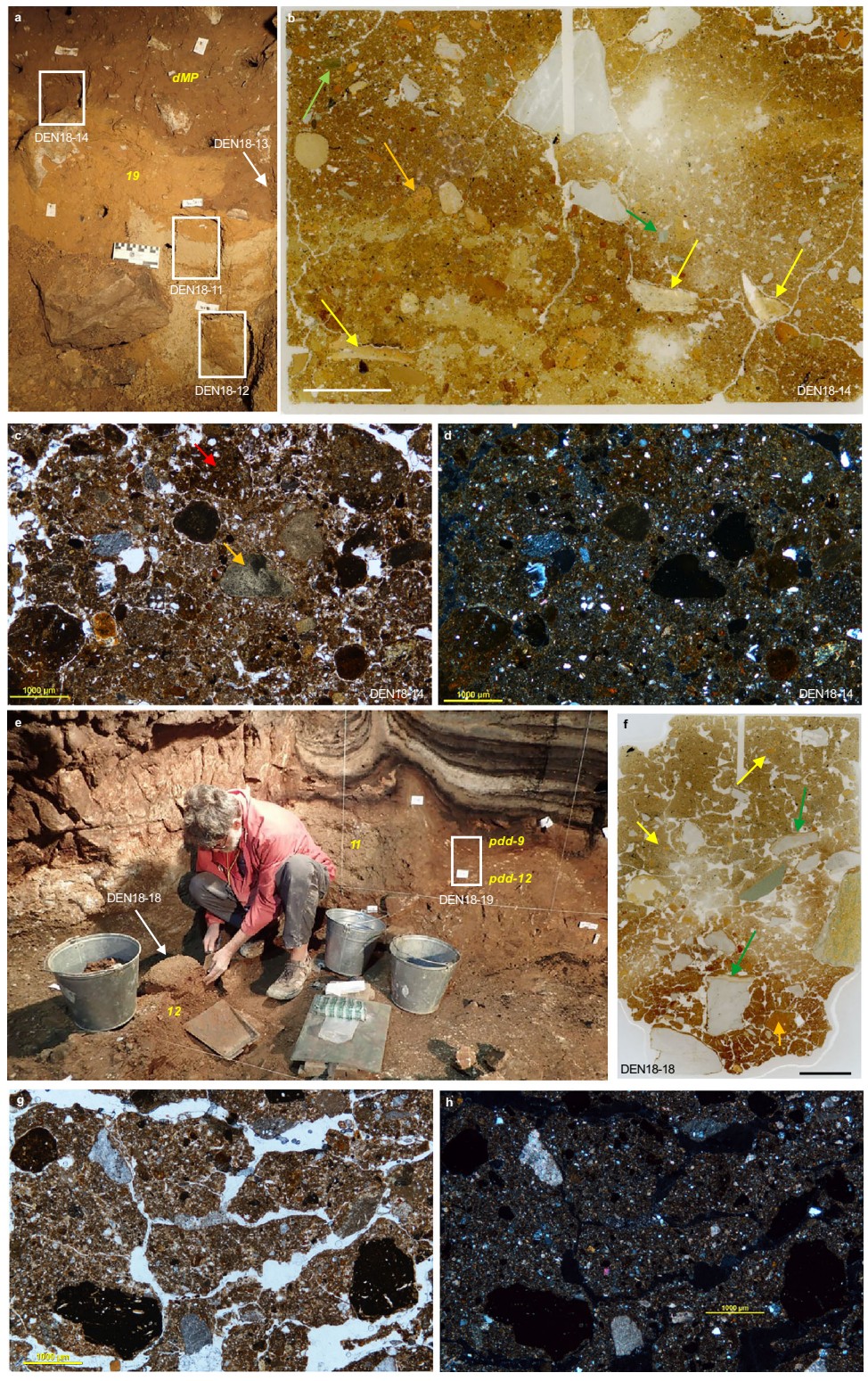

contains more than 70% of the grains (Supplementary Data 9). We consider the ages determined for 31 of these samples to be the most reliable estimates of the time of deposition of the associated layers, and used them to develop a Bayesian model of the chronology of the stratigraphic sequence in South Chamber (model A; Fig. 5, Supplementary Fig. 23 and Supplementary Code 1). The resulting age estimates are cited below and displayed in Supplementary Fig. 25d, with total age uncertainties at the 95% confidence interval (CI). We included all 50 ages in an alternative Bayesian age model to further constrain the

start and end ages of the archaeological phases (model B; Supplementary Figs. 24, 25e and Supplementary Code 2). The two models yield statistically indistinguishable ages.

Samples from Profiles A, B and C commonly have multi-component distributions with fewer than 70% of the grains in any component, reflecting the incorporation of a substantial proportion of grains associated with burrowing or redeposited clasts of older sediment. For example, samples from layer 11 typically consist of grains that fall into two discrete age clusters (discussed below), and samples

**Fig. 3 | Microstratigraphy (micromorphology) of sediment blocks from South Chamber. a** Locations of sediment blocks DEN18-11, −12 and −14 (white rectangles) in the lower southeast face of Profile A showing the unconformable contact between layer 19 (yellowish-brown silty clay with grey patches due to gleying) and the light brown dMP deposits. Stratigraphic layers are indicated in yellow text. DEN18-13 (white arrow) was collected from the upper part of layer 19 and lower part of the dMP deposits in the adjacent southwest face, 30 cm to the right of DEN18-11 (Supplementary Fig. 1a). Sediment blocks were collected, prepared for analysis, examined with stereoscopic and petrographic microscopes at high magnification, and the micromorphological features described using procedures documented previously[27]. **b** Thin-section scan of DEN18-14 showing a mixture of bioturbated material including bone (yellow arrows), coprolite (orange arrow) and rock clasts (green arrows) in the lowermost dMP deposits, close to the contact with layer 19 (scale, 1 cm). **c** Thin-section photomicrograph in plane-polarized light (PPL) of DEN18-14 showing a burrowed mixture of rounded components, including mostly brown slightly sandy, silty clayey aggregates (red arrow) and some coprolites (orange arrow) (scale, 1 mm). **d** Same as (**c**) but in cross-polarized light (XPL) (scale, 1 mm). **e** Excavation of DEN18-18 (white arrow) from the reddish-brown sediments (layer 12) between Profiles B and C. DEN18-19 in Profile C (white rectangle) spans pdd-9 and −12 (Supplementary Fig. 2a). Stratigraphic layers are indicated in yellow text. DEN19-1 was collected from the deposits adjacent to Denisova 22 and spans the upper part of layer 19 and the overlying sediments (layer 13) (Supplementary Fig. 3a). **f** Thin-section scan of DEN18-18 showing slightly platy and rounded aggregated microstructures typical of freeze–thaw processes, phosphatized rims on limestone clasts (green arrows), fine bone fragments (yellow arrows) and a coprolite (orange arrow) (scale, 1 cm). **g** Thin-section photomicrograph (in PPL) of DEN18-18 showing a platy structure developed in compact silty clay that contains various types of rock clasts (scale, 1 mm). The generally well-sorted fine fraction has a loess-like aspect, although its origin is uncertain. **h** Same as (**g**) but in XPL (scale, 1 mm).

DCS18-8 and DCS18-11 intersected burrows filled with Holocene-age sediments. Yet despite these complications, individual components often have ages that closely match those of samples from the same or adjacent layers that have distributions dominated by a single component or one containing more than 70% of the grains (Fig. 4).

Layer 19 accumulated before $247 \pm 39$ ka and layer 17 was deposited $202 \pm 35$ to $167 \pm 29$ ka. The latter layer was exposed only on the left side of Profile D when sampled for dating and is considered stratigraphically intact, albeit deformed, based on field observations (Fig. 2d and Supplementary Fig. 26c) and the single-component $D_e$ distribution for sample DCS19-13.

The dMP samples ($n = 18$) have ages that range from ~180 to ~20 ka for individual components, but 14 of these samples (78%) have distributions dominated by a single population of grains or have a component that contains more than 70% of the grains; we included these samples in age model B (Supplementary Figs. 24, 25e). Notably, the dMP ages cluster spatially into three broad groups (140–120, 90–60 and 50–20 ka) separated by steeply dipping contacts (Supplementary Fig. 22f). This pattern closely matches the stratigraphy clearly expressed on the left side of Profile D (Supplementary Fig. 22d), which is located 1–2 m further into the cave (Fig. 1). Thus, despite the lithostratigraphic difficulties in distinguishing layers within the dMP deposits and the micromorphological evidence for incorporation of reworked materials, some stratigraphic integrity appears to have been retained, demonstrating the value of single-grain optical dating for gaining insights into complex site-formation histories.

Layers 16 and 15 were deposited $148 \pm 22$ to $120 \pm 15$ ka (based on four samples with single-component distributions and statistically consistent OSL and pIRIR ages) and, following a modelled time gap, layer 14 accumulated $106 \pm 17$ to $86 \pm 12$ ka. Although the ages for the two samples from this layer differ by ~20 ka (Supplementary Fig. 22d), they each have single-component distributions and concordant OSL and pIRIR ages. Layer 13 was deposited $79 \pm 11$ to $67 \pm 7$ ka, with the much older pIRIR age component in samples DCS19-1a and DCS19-1b reflecting the micromorphological evidence for redeposition of sediment reworked from layer 19 (Supplementary Fig. 3).

The uppermost layers of the Pleistocene sequence in South Chamber have been tentatively divided into layer 12, pdd-12, layer 11 and pdd-9[1,3]. The most intact and least phosphatized parts of layer 12 have modelled start and end ages of $65 \pm 7$ to $56 \pm 7$ ka, based on two samples each in Profiles C (left-hand side) and D. Given the complex formation and post-depositional history of pdd-9 and −12, we treated all samples from these deposits (henceforth pdd-9/12) as an undifferentiated group in the age model and obtained a wide age range of $52 \pm 7$ to $28 \pm 5$ ka.

All six samples from layer 11 have multi-component distributions, with ages of approximately 80–40 ka and 30–20 ka for the two main components. We attribute the older component to the incorporation of grains into layer 11 from pdd-9/12 and/or layer 13, possibly during the formation of pdd-9. The $^{14}C$ ages for layer 11 are similarly spread[13,31], with three ages older than 49 ka BP and three younger ages (39.2–37.6, 34.5–32.3 and 24.2–23.8 cal. ka BP, 95% CI). Only one sample from this layer (DCS17-4) has more than 70% of both quartz and K-feldspar grains in the main component. The modelled start and end ages of $25 \pm 5$ and $18 \pm 6$ ka for layer 11 encompass the youngest $^{14}C$ age and the genetic age estimates of approximately 24.7 and 18.5 ka for the pendant[31] from the upper part of this layer. All three dating methods, therefore, indicate that parts of layer 11 accumulated approximately 25–20 ka, with older materials reflecting a more complex history of formation that includes the syn- or post-depositional incorporation of reworked material.

Layers 7 and 5 accumulated from $14 \pm 6$ ka, based on two samples that have single-component $D_e$ distributions and statistically consistent OSL and pIRIR ages. The underlying layer (layer 8) is thin, organic-rich and intersected by burrows. We infer a depositional age of $19 \pm 4$ to $15 \pm 4$ ka for this layer, based on the inclusion of sample DCS18-8 in model B. Most of the grains in this sample are from a burrow and have ages consistent with the weighted mean age of $5.3 \pm 0.5$ ka for the overlying sample (DCS18-7) from layer 5. The latter age accords with the $^{14}C$ chronology for the Afanasievo culture (5.3–4.8 cal. ka BP, 95% CI)[62,63] associated with pottery from this layer[64–66].

## Sediment DNA of ancient hominins

Screening of mammalian and hominin mtDNA preservation yielded evidence for the presence of ancient faunal and hominin DNA in 326 (75%) and 56 (13%), respectively, of the 437 sediment samples analyzed from South Chamber (Fig. 6; Supplementary Fig. 27; Supplementary Section 3). These proportions are lower than in Main and East Chambers (94% and 30% for ancient mammalian and hominin mtDNA, respectively), due partly to the scarcity of DNA recovered from samples in the phosphatized and weakly acidic parts of pdd-9/12, characteristics known to be detrimental to DNA preservation[3].

The oldest hominin DNA, recovered from three samples in layer 17 (one on the border with layer 19), is of Denisovan origin and associated with early Middle Palaeolithic artefacts (Fig. 6d). A $k$-mer-based approach[3,21,67] revealed affinities between the mtDNA fragments from one of these samples with Denisova 2 and Denisova 8 (Fig. 7; Supplementary Fig. 30), consistent with the occurrence of these Denisovan mtDNA lineages in layers older than 150 ka in Main and East Chambers[3]. Notably, recent mtDNA analysis of Denisova 13 (unpublished) shows that the mitochondrial genome of this skull fragment is also more similar to the genomes of Denisova 2 and Denisova 8 than to those of Denisova 3 and Denisova 4, which are younger. Based on our sediment DNA results for the oldest hominins, we tentatively

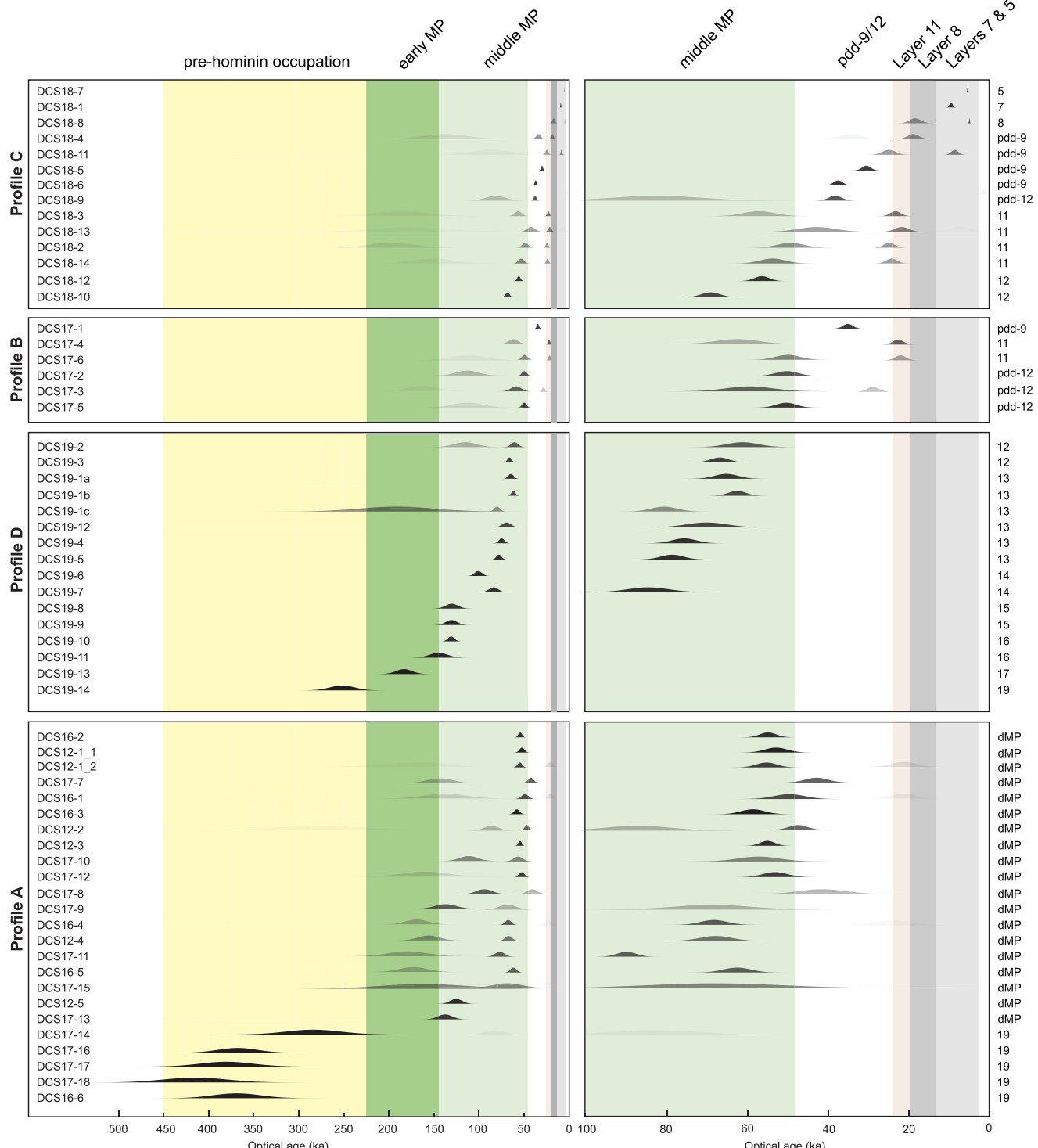

**Fig. 4 | Optical ages of sediment samples from South Chamber.** Age distributions for all samples (*n* = 60). The left and right panels extend to 500 ka and 100 ka, respectively, with samples arranged by stratigraphic sequence (Profiles A, D, B and C, bottom to top). Sample codes are shown on the left and the corresponding layers on the right. Archaeological phases are denoted by coloured shading (early Middle Palaeolithic, dark green; middle Middle Palaeolithic, light green; Upper Palaeolithic, light orange) and the labels at the top; the period prior to hominin occupation is shaded light yellow. The width of each normal distribution reflects the age uncertainty (standard error of the mean for the total random (i.e., unshared) components of error); their heights are identical. The relative proportion of grains in individual age components is indicated by density of shading on a grey scale ranging from 0% (transparent) to 100% (black). Darkest shades correspond to samples with single-component distributions and samples that have multiple-component distributions with more than 70% of grains in the main component. Lightest shades denote components that contain fewer than 10% of grains.

assign Denisova 13 to layers 17 or 18. An age of more than 150 ka is congruent with the presumed origin of this fossil in the older part of the stratigraphic sequence[2,20,24]. A Denisovan molar (Denisova 25)[25] has since been recovered from layer 17.

The next oldest layers with ancient hominin mtDNA are middle Middle Palaeolithic layers 16–12 on the right-hand side of Profile D (Fig. 6d), where nine of the 10 optical dating samples have single-component $D_e$ distributions, providing confidence in the

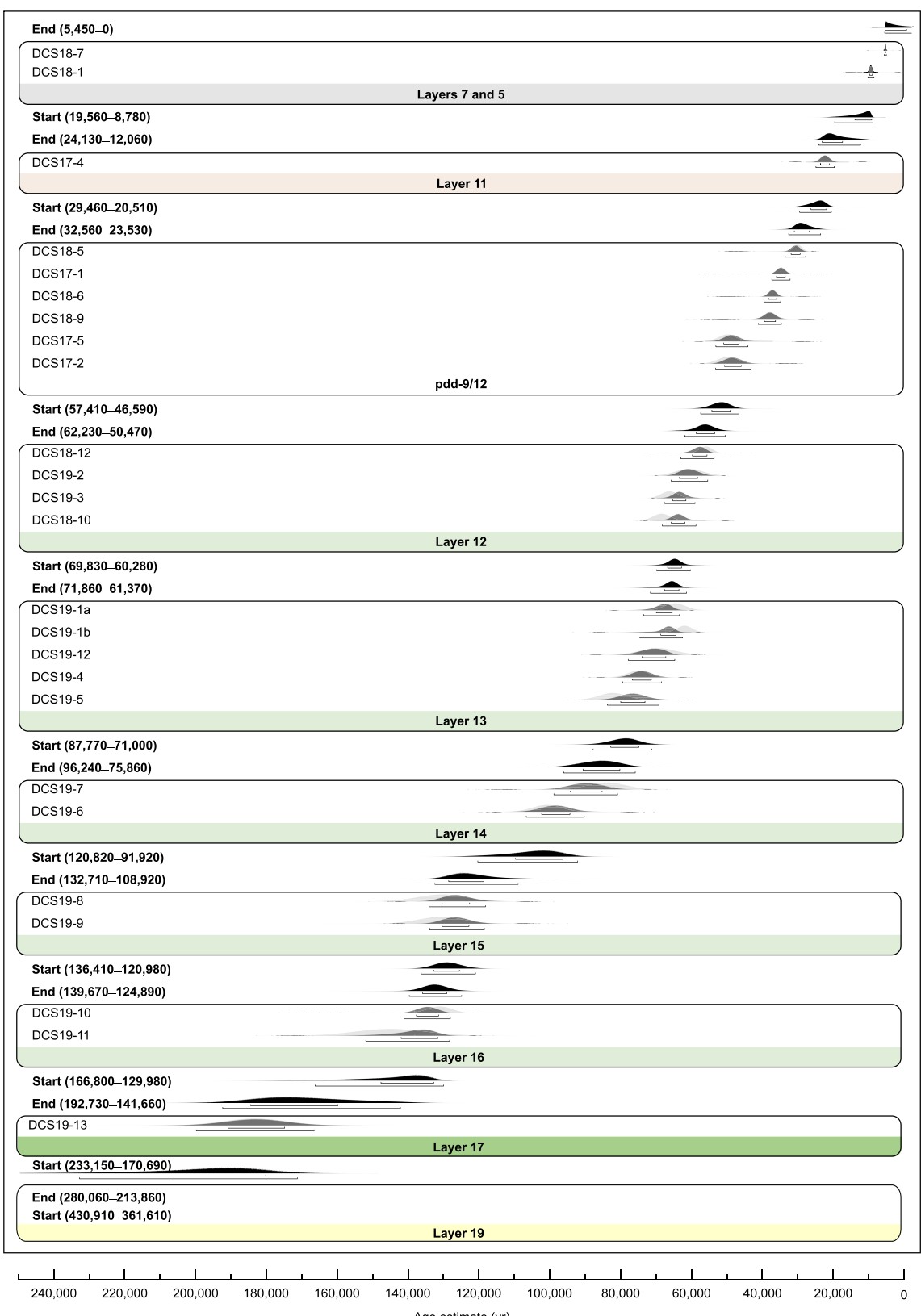

stratigraphic integrity of these layers. Of the 27 samples that yielded ancient hominin mtDNA, 24 could be assigned to either Neanderthals or Denisovans (Fig. 6d).

Two samples from each of layers 16 and 15 yielded hominin mtDNA, one of which we identified as Denisovan (layer 16) and the other as Neanderthal (layer 15). These layers accumulated 148 ± 22

to 121 ± 15 ka, spanning the climatic transition between the penultimate glacial and last interglacial periods. Denisovan DNA was retrieved from two samples in layer 14, which has the fewest artefacts of any layer in South Chamber (Supplementary Data 2), and from 17 samples in layer 13. Three of the four samples dated from these layers have single-component $D_e$ distributions, so we consider the sediment DNA to be in

**Fig. 5 | Bayesian age model A for South Chamber.** Ages included in this model ($n = 31$) are considered the most reliable estimates of time of deposition of the relevant layers; they correspond to samples with single-component $D_e$ distributions or multi-component distributions with more than 70% of grains in the main component (ages shown in bold font in Supplementary Fig. 22). Individual sample codes are shown on the left, together with the modelled start and end age-ranges in years (95% confidence interval, CI) for each layer or combination of layers (colours as in Fig. 4), rounded off to the nearest decade; the corresponding probability distributions are shown in black. Probability distributions for measured ages (likelihoods) and modelled ages (posterior probabilities) of individual samples are shown in light and dark grey, respectively. The narrow and wide brackets beneath each distribution denote the 68% and 95% CIs, respectively. To show details for samples younger than 200 ka, the age distributions of samples from layer 19 ($n = 6$) are not displayed here, but are included in the model; all age distributions are displayed in Supplementary Fig. 23. The corresponding results for samples included in Bayesian age model B ($n = 50$) are shown in Supplementary Fig. 24. For both models, the ages and associated total random (i.e., unshared) components of error were modelled using OxCal version 4.4.4.

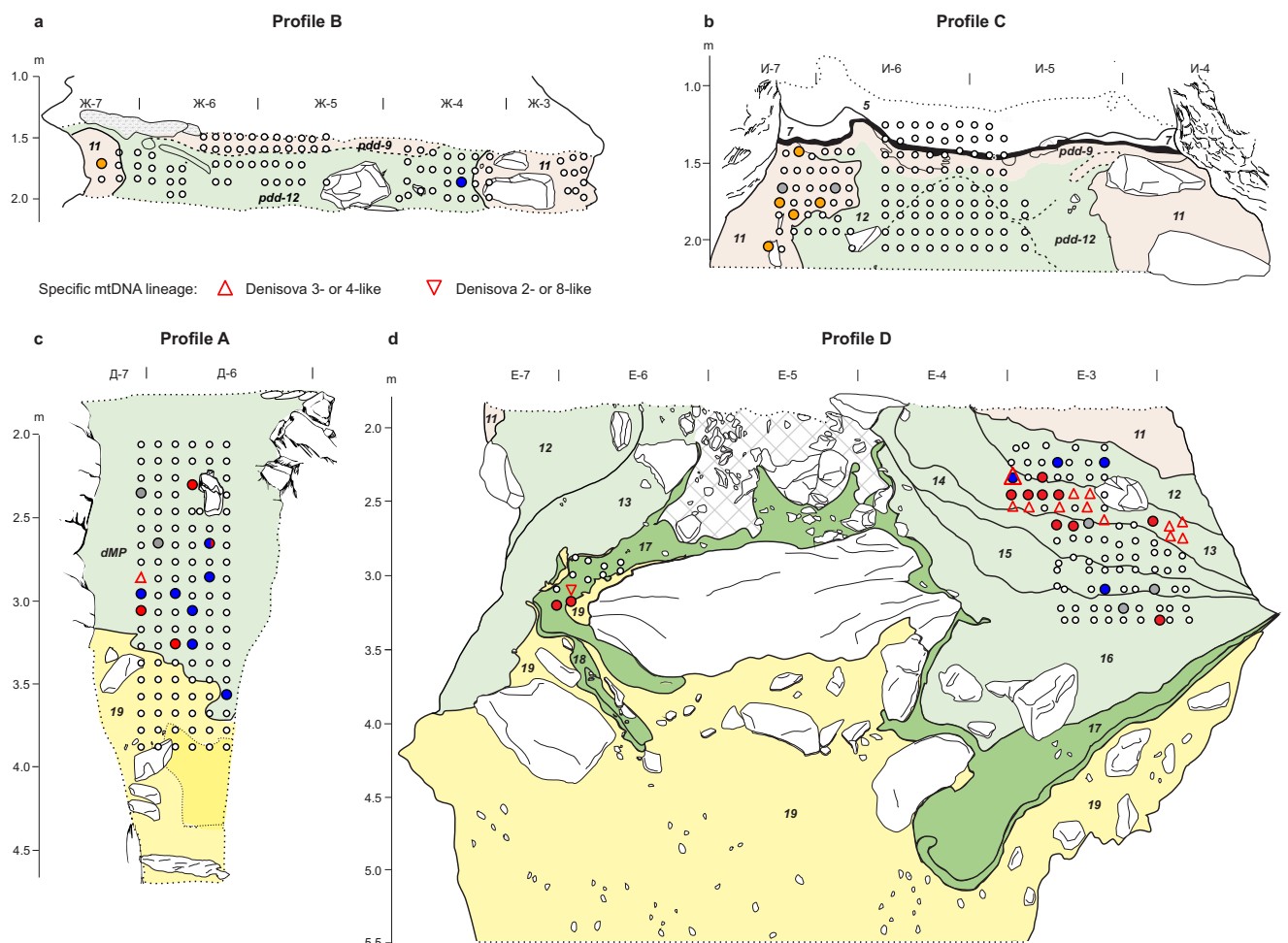

**Fig. 6 | Ancient hominin DNA in sediment samples from South Chamber. a–d** Results for samples from Profiles B, C, A, and D, respectively; small rocks have been removed for clarity. Layer colours denote archaeological phases, as in Fig. 4. Layer 19 is shaded light yellow and areas in white as in Fig. 2. Filled circles indicate locations of sediment samples analyzed for mtDNA and colours correspond to hominin mtDNA detected: Denisovan (red), Neanderthal (blue), ancient modern human (dark yellow), unidentified ancient hominin (grey) and no ancient hominins detected (white). Other symbols denote samples for which mtDNA could be assigned to a specific Denisovan lineage: Denisova 3- or 4-like (red triangles) or Denisova 2- or 8-like (red inverted triangles). Denisovan mtDNA was also retrieved from one of three sediment samples associated with Denisova 22 (layer 13). Ancient modern human mtDNA was recovered from clumps of sediment attached to the pendant and from beneath the figurine (both layer 11). Data for Profiles A and B are from[3].

secure stratigraphic context and deposited $106 \pm 17$ to $67 \pm 7$ ka. Denisovan mtDNA was also recovered from one of the samples collected from the layer 13 sediments beneath Denisova 22; the optical dating samples closest to this fossil have ages of $65 \pm 4$ and $62 \pm 3$ ka (Supplementary Fig. 22c).

Eleven of the mtDNA samples from layer 13 show affinities to Denisova 3- or 4-like sequences (Fig. 7; Supplementary Fig. 30), consistent with the identification of this lineage in sediments younger than ~80 ka in Main and East Chambers[3] and at Baishiya Karst Cave in Tibet[68]. This lineage may have appeared earlier (~100 ka) based on its detection in one of the layer 14 samples, but we note that the latter is from close to the contact with layer 13. Neanderthal mtDNA was retrieved from one sample in layer 13 and two in layer 12 (deposited $65 \pm 7$ to $56 \pm 7$ ka). One of the optical dating samples from layer 12 (DCS19-2) has an older component (Fig. 4), so we cannot discount the possibility that the Neanderthal mtDNA originates from the reworked material.

Denisovan, Neanderthal and unclassified ancient hominin mtDNA has previously been retrieved from the dMP deposits[3] (Fig. 6c). As these deposits consist of mixed-age sediments, we cannot reliably infer temporal patterns of Denisovan and Neanderthal presence from the sediment DNA data. We note, however, that these data broadly support the sediment DNA patterns in Profile D, with the identification of

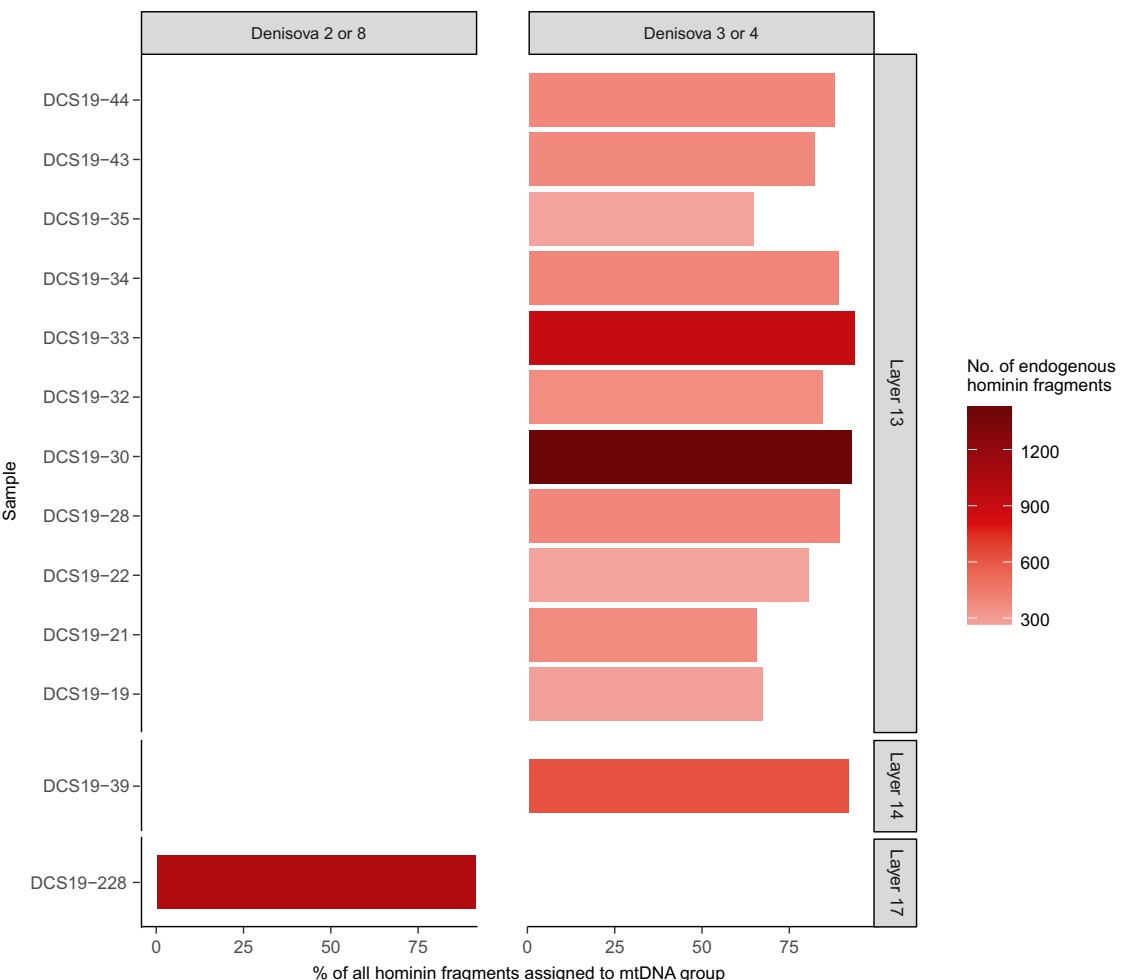

**Fig. 7 | Denisovan mtDNA lineages identified in sediment samples from South Chamber.** Specific lineages were identified using kallisto[67] (see "Methods").
**a** Affinities to Denisova 2, Denisova 8, Denisova 3 or Denisova 4 of samples in Profile D for which hominin mtDNA fragments could be assigned to a specific Denisovan mtDNA lineage (Fig. 6). Bars are colour-coded by the number of endogenous hominin mtDNA fragments. Sample codes are shown on the left and the corresponding layers on the right. Supplementary Fig. 30 shows the assignments for each mtDNA group.

Denisova 3- or 4-like mtDNA in sample S127 on the left side of Profile A[3] being consistent with an age of less than 80 ka for the dated sediments (Supplementary Fig. 22f).

Only one of 164 samples (<1%) from pdd-9/12 and the Holocene layers yielded ancient hominin DNA (Fig. 6a, b), presumably due to the abundance of phosphates[1,3,32,33]. Neanderthal DNA was retrieved from this sample (S55 in ref. [3]). Although 87% of grains in the nearest dated sample (DCS17-5) were deposited $50 \pm 6$ ka, we cannot exclude the possibility that the mtDNA is associated with the older component ($113 \pm 17$ ka).

Ancient modern human DNA was recovered from eight of 52 samples (15%) from layer 11: six from Profiles B and C (Fig. 6a, b) and two from the pendant and figurine. As none of the samples yielded significant evidence of Denisovan or Neanderthal DNA, despite layer 11 containing grains older than 50 ka (Fig. 4), we consider the mtDNA to be mainly associated with the sediments deposited approximately 25–20 ka. This age range is consistent with genetic age estimates for the pendant from which ancient modern human DNA was recovered and with the [14]C age of the charcoal sample closest to it[31], clarifying the chronological attribution of this artefact[69]. The sediment DNA samples likely correspond, therefore, to the Upper Palaeolithic parts of layer 11, rather than those associated with older Upper Palaeolithic artefacts[36–47,50–52].

A Denisovan molar (Denisova 4) was recovered from layer 11 on the same side of the chamber as the pendant and figurine, but ~2 m closer to the entrance and only 5 cm above the contact with layer 12[13]. It seems likely, therefore, that Denisova 4 was reworked from layer 12 into layer 11 with the 60–50 ka sedimentary component identified in samples DCS17-6, DCS18-2 and DCS18-3 on the right side of the ridge. This scenario is congruent with the modelled age of 84–55 ka for Denisova 4 estimated using genetic and stratigraphic information[2], the [14]C ages older than 49 ka BP obtained for animal bones from layer 11[13], and the optical ages of $66 \pm 4$ and $61 \pm 5$ ka for the layer 12 samples in Profile D (Supplementary Fig. 22d).

The sediment DNA results for ancient hominins in all three chambers of Denisova Cave are shown in relation to the Middle and Upper Palaeolithic archaeological phases and the modelled age estimates for Denisovan and Neanderthal fossils[2,22] in Fig. 8.

## Sediment DNA of ancient mammals

The layer 19 samples are dominated by the mtDNA of ursids (cave bear and brown bear in similar proportions) and, to a lesser extent, hyaenids (cave hyaena) and bovids (e.g., bison, ibex) (Fig. 9b; Supplementary Fig. 27 in [3]). Five samples from a gleyed patch in Profile A yielded no ancient DNA[3], presumably due to the anaerobic conditions. The mtDNA of canids (e.g., wolf, red fox, dhole) dominates samples from layer 17 and some have substantial proportions of

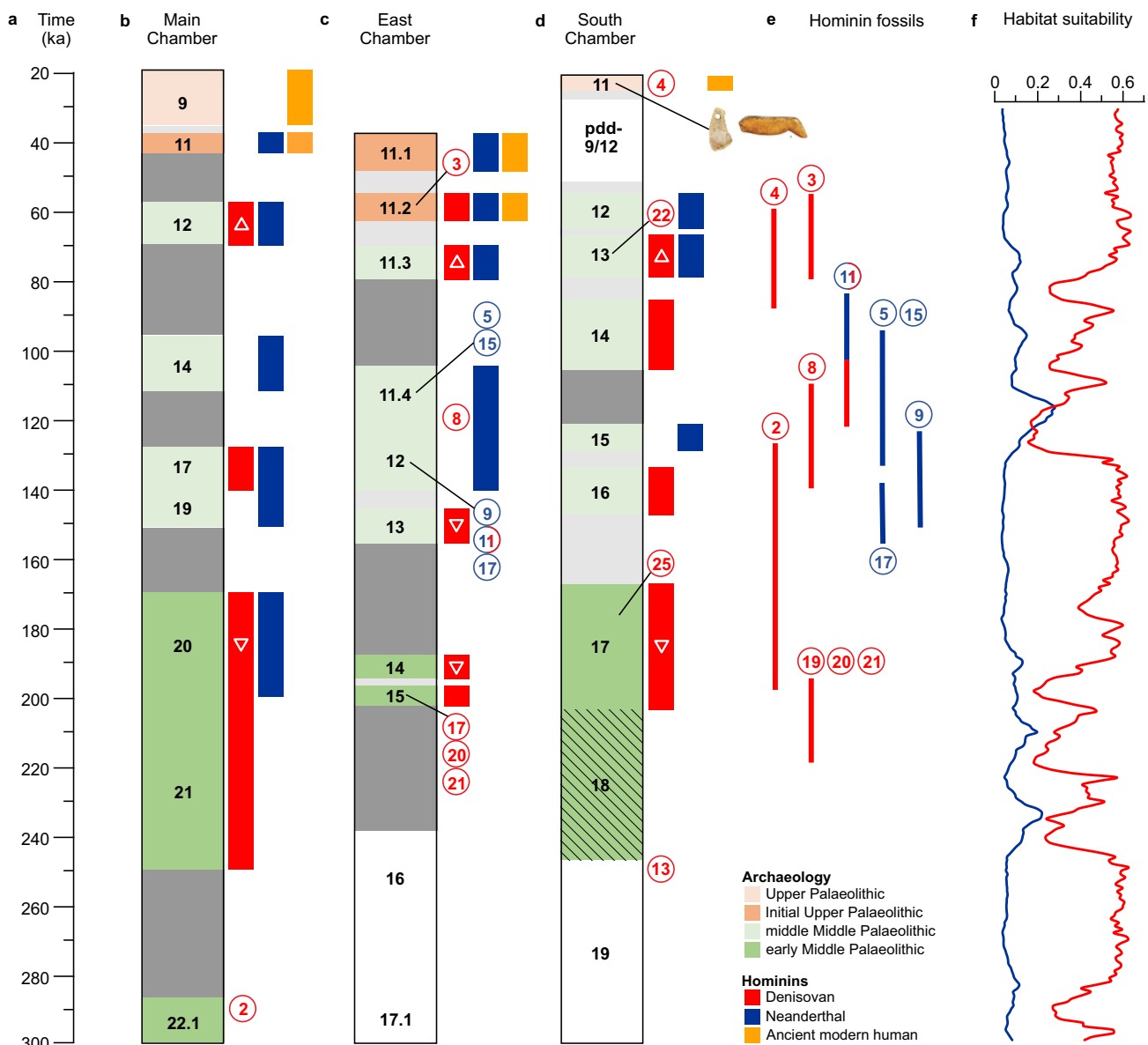

**Fig. 8 | Timeline of ancient hominins and archaeological phases at Denisova Cave. a** Common time scale for all three chambers. **b–d** Pleistocene stratigraphic sequences in Main, East and South Chambers, respectively. Chronological, archaeological and genetic data for Main and East Chambers from[1,3]. The South Chamber chronology is based on Bayesian age model A (Supplementary Fig. 25d); layer 18 was not sampled for optical dating or sediment DNA. Colours denote archaeological phases, as in Figs. 4–6. Initial Upper Palaeolithic layers in Main and East Chambers are shaded dark orange. Layers with no archaeology and pdd-9/12 are shown in white. Modelled time gaps are shaded dark grey and time gaps between point estimates of age for successive layers with overlapping age uncertainties are shaded light grey. Other colours denote hominin mtDNA detected in sediments: Denisovan (red), Neanderthal (blue) and ancient modern human (dark yellow). Symbols for specific Denisovan mtDNA lineages inset in white: Denisova 3- or 4-like (triangles) or Denisova 2- or 8-like (inverted triangles). Circled

numbers indicate individual fossils of Denisovans (red), Neanderthals (blue), the Neanderthal–Denisovan offspring (both colours) and the layers from which they were recovered, as originally reported. We now consider Denisova 8 was most likely retrieved from layer 14 in East Chamber, and Denisova 4 was likely reworked from layer 12 in South Chamber. The relative stratigraphic positions of the pendant and figurine are also shown in (**d**). **e** Age ranges (95% highest posterior density) for fossils of Denisovans, Neanderthals and the Neanderthal–Denisovan offspring, estimated using a Bayesian model that combines chronometric and stratigraphic information with data from hominin DNA sequences[2,22]. **f** Habitat suitability for Denisovans (red) and Neanderthals (blue) in the Altai region[70]. Simulated data are the average for two species distribution models in 1000-year timesteps and can be interpreted in terms of probability of hominin presence, ranging from 0 (habitat not suitable) to 1 (habitat extremely suitable).

ursid DNA, mainly of brown bear (Fig. 9b; Supplementary Figs. 27c, 28b, 29).

Samples from layers 16 and 15 contain mtDNA from a wide variety of large mammals, including hyaenids, cervids (e.g., deer, elk), equids (horse), woolly mammoth and woolly rhinoceros (Fig. 9b; Supplementary Figs. 27b, 28f). The proportion of bovid DNA generally increases from layers 14 to 12, but fragments of cervid, hyaenid, equid

and woolly rhinoceros DNA also occur in relatively high proportions in several of the layer 12 samples. We identify a shift in hyaenid mtDNA from haplogroup D in layer 17 to haplogroup A in all younger layers (Supplementary Figs. 28d–f, 29).

The dMP deposits are dominated by hyaenid and bovid DNA, and several samples contain substantial proportions of cervid and equid DNA (Supplementary Fig. 27 in ref. 3). The high proportion of ursid

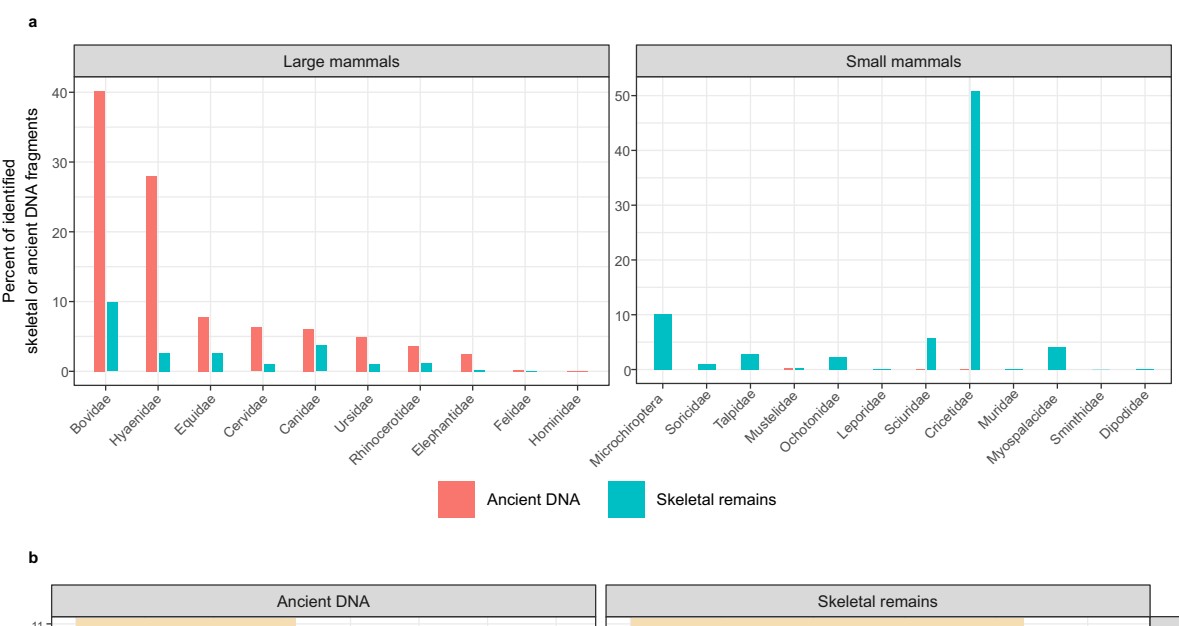

**Fig. 9 | Proportions of mammalian mtDNA fragments and skeletal remains in South Chamber. a** Proportions (in percent) of ancient DNA fragments (red) and skeletal remains (blue) assigned to taxonomic families (or the suborder Micro-chiroptera) of large and small mammals, combined for all samples and ranked in descending order of mtDNA percentage. **b** Proportions (in percent) of ancient mtDNA fragments and skeletal remains assigned to families of large mammals for individual layers, arranged in relative stratigraphic order. The mtDNA data were obtained by averaging across the percentages of fragments assigned to each family in all samples from a layer.

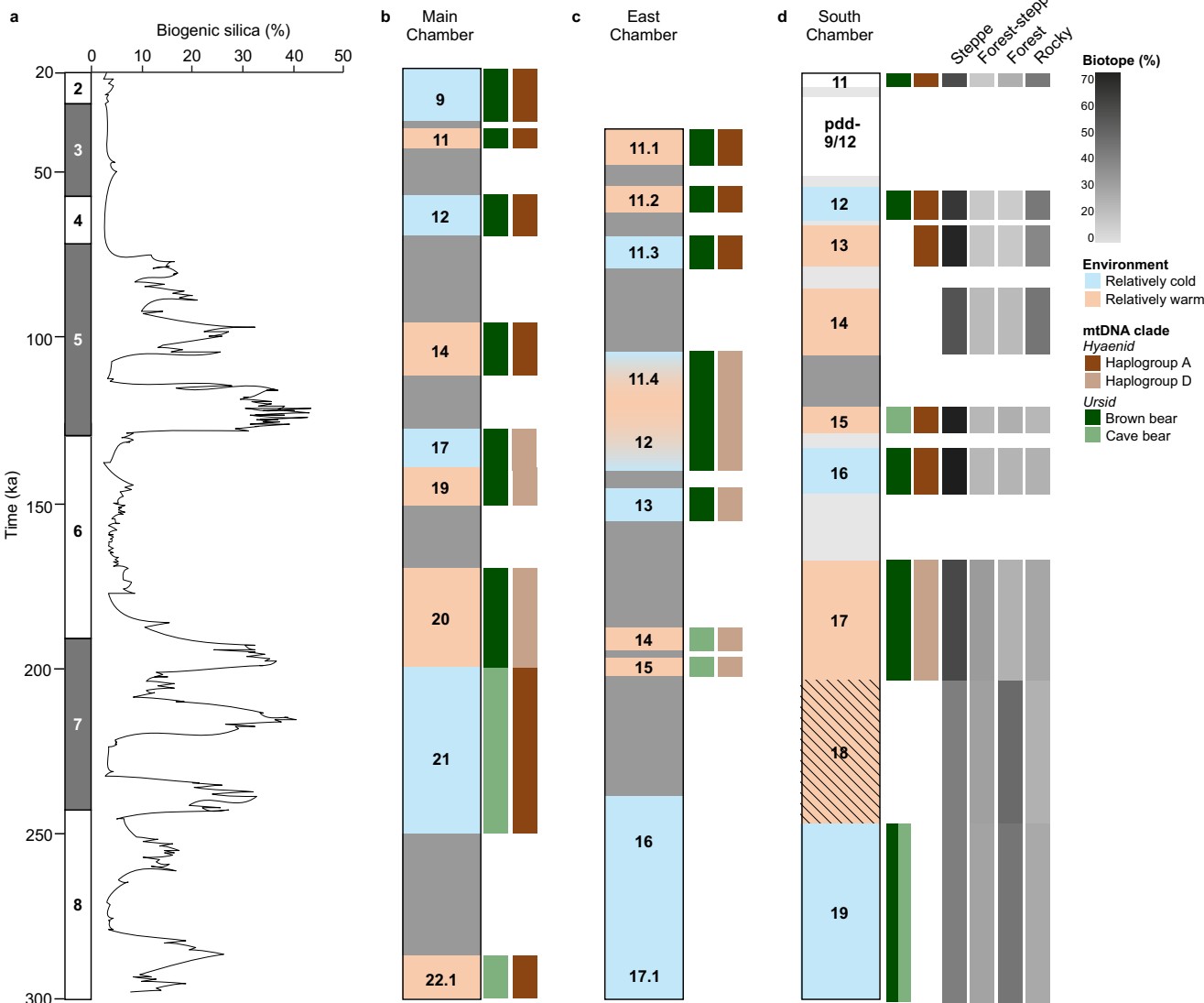

**Fig. 10 | Comparison of Pleistocene faunal and environmental records at Denisova Cave. a** Composite biogenic silica record of diatom productivity in Lake Baikal, a proxy for regional annual temperature[98], as a function of time. Boundary ages of marine isotope stages 8 to 2 from[99]. **b–d** Stratigraphic sequences in Main, East and South Chambers, respectively. Ages for layers and time gaps as in Fig. 8 and Supplementary Fig. 25d. Local climatic conditions relative to the current climate (moderately continental) are inferred from the skeletal remains of small vertebrates (orange, warm; blue, cold). Conditions inferred for pdd-9/12 and layer 11 from the remains of large and small fauna are inconsistent, reflecting the complex formation history of these layers (white). Other colours denote mtDNA data for dominant ursid and hyaenid populations detected in sediments: brown bear (dark green), cave bear (light green), cave hyaena haplogroup A (dark brown) and cave hyaena haplogroup D (light brown). Chronological, climatic and genetic data for Main and East Chambers from[1,3]. Layer 19 in South Chamber has approximately equal proportions of cave and brown bear (Supplementary Fig. 29). Relative proportions of skeletal remains of large mammals associated with steppe, forest-steppe, forest and rocky habitats (biotopes; Supplementary Data 4) are displayed on a grey scale in **d**.

DNA in samples closest to the contact with layer 19 is likely due to the incorporation of reworked material from the latter. No other spatial patterns in faunal composition are evident in the dMP deposits, as expected given the incorporation of reworked sediments in many of these samples.

Ancient faunal DNA was recovered from only 66 of 166 (40%) samples from the Holocene deposits and extensively phosphatized parts of pdd-9/12. The least phosphatized parts of pdd-9/12, the layer 13 sediments beneath Denisova 22, and the layer 11 sediments in Profile B and beside the pendant, are dominated by hyaenid and bovid DNA, with smaller proportions of equid, canid and ursid (brown bear) DNA (Fig. 9b; Supplementary Fig. 27a, b; Supplementary Fig. 27 in ref. 3). The latter pair dominate the layer 11 sediments in Profile C and beside the figurine, perhaps because these areas are closest to the cave walls and served as dens for bears, wolves or red foxes.

The large mammals identified in the sediment DNA samples are also represented in the fossil record (Fig. 9; Supplementary Section 1). For layers to which we can confidently assign numbers, the proportions of mammalian mtDNA fragments and skeletal remains assigned to families of large mammals are broadly consistent, despite the fragmentary nature of the fossils and differences in the amount of DNA deposited by different taxa. The fossil record of small mammals (Supplementary Section 1), however, is not reflected in the genetic data (Fig. 9a). This may be an experimental artefact (underrepresentation in the mtDNA capture probes) or the result of large mammals having a greater biomass; we observed similar patterns in Main and East Chambers[3]. More generally, we caution that patterns inferred for ancient fauna using sediment DNA are subject to the same uncertainties as for ancient hominins, especially in places where

sediments consist of multiple age components resulting from the incorporation of reworked material[60].

Palaeontological data for the faunal assemblages (more than 130,000 specimens of at least 43 species of large mammals[53,54,56,58], and more than 30,000 specimens from 46 taxa of small mammals, birds, reptiles, amphibians and fish[55,57,59]) allow us to reconstruct the environmental conditions associated with layers 19–11 in South Chamber (Fig. 10d; Supplementary Section 1). The broad patterns inferred from the remains of small vertebrates indicate that layer 19 accumulated under relatively cold conditions with open landscapes, and steppe and forest habitats dominated during the relatively warm penultimate interglacial (layers 18 and 17). Forest cover decreased during the subsequent relatively cold period (layer 16) and then expanded, along with grass meadows, during relatively warm and more humid conditions (layers 15–13). A period of cooler climate favoured the spread of steppe habitat and reduction in forest cover (layer 12), followed by an interval of relatively warm and dry climate dominated by forest and steppe landscapes (layer 11). The remains of large mammals likewise indicate a mosaic landscape, with rocky habitats persisting throughout the time of deposition of layers 19–11 and forests alternating with patches of meadow and steppe.

Major turnovers of the Pleistocene fauna in Main, East and South Chambers are most evident between ~200 and ~170 ka and between ~130 and ~100 ka, which are broadly synchronous with climatic transitions from an interglacial period to a glacial period (Fig. 10; Fig. 2 in ref. 3). Differences between the Lake Baikal and Denisova Cave climatic records are likely due to differences in scale (diatom productivity is a proxy for regional annual temperature, whereas environmental conditions reconstructed from faunal and pollen records reflect changes in the local ecology) and to factors that affect interpretation of the cave records[1,3]. Differences between the environmental records inferred for the three chambers reflect their complex stratigraphic sequences and individual histories of sediment accumulation, erosion and post-depositional modification. Each chamber is, in effect, a discrete site with a unique sedimentary record. We also note that the reconstructed environmental conditions for Main and East Chambers are based on both pollen and faunal records, whereas only the latter data are currently available for South Chamber.

Comparisons of the Lake Baikal and Denisova Cave records are also limited by the time-averaging associated with the accumulation of the cave sediments and by the disturbance of parts of some layers by post-depositional deformation and burrowing (mainly the uppermost Pleistocene layers in East and South Chambers, and the dMP deposits in the latter). In addition, the optical ages have uncertainties that are too imprecise to resolve time gaps or climatic fluctuations of several millennia within the stratigraphic sequences, and the ages of some layers are poorly constrained (e.g., layers 14 and 17 in South Chamber). Given these caveats, the overall consistency of the archaeological, hominin, faunal and climatic records across all three chambers is striking.

## Discussion

The data for South Chamber largely support the current chronology for occupation of Denisova Cave by archaic hominins and ancient modern humans, and the relative timing of changes in hominin populations, faunal diversity and environmental conditions[1–3,22,29] (Figs. 8, 10). The sediment DNA results for South Chamber fill most of the time gaps in the Main and East Chamber records, and more than double the number of Denisovan mtDNA sequences associated with the latest Middle Palaeolithic layers (deposited approximately 80–50 ka). They also augment the genetic evidence of modern humans at Denisova Cave between ~25 and ~20 ka. This Upper Palaeolithic population may be related to the Ancient North Eurasian maker or wearer of the pendant of similar age[31].

A whole-of-cave synthesis of the chronological, sediment DNA, archaeological and faunal data reveals several broad patterns over the past three glacial–interglacial cycles, albeit with the caveats noted above. Of the 963 sediment DNA samples analyzed, ancient hominin and faunal mtDNA was identified in 213 (22%) and 845 (88%) samples, respectively. The oldest layers with Denisovan mtDNA accumulated from ~250 ka and the oldest Denisovan fossils (including Denisova 25 from which a high-coverage genome has been reconstructed[25]) were deposited ~200 ka, both in association with early Middle Palaeolithic stone tools (Fig. 8). In East Chamber, the earliest Denisovan deposits (layer 15, ~200 ka) and the thin, underlying layer (layer 16), contain microscopic traces of fire use (e.g., melted phytoliths, bands of dispersed charcoal, fragments of crushed burnt bone, possible phosphatized ashes), but intact combustion features are absent (Supplementary Fig. 4 and ref. 27).

Neanderthals first appeared shortly after 200 ka, but many of the samples that yielded Neanderthal DNA and all four fossils of Neanderthals and Denisova 11 (Fig. 8e) were recovered from middle Middle Palaeolithic layers deposited approximately 150–80 ka. We retrieved Denisovan DNA from only a few sediment samples in this interval, and only one Denisovan fossil has been reported from layers of this age (Denisova 8, which we now consider was most likely recovered from layer 14 in East Chamber, ~190 ka, based on our reassessment of its stratigraphic context).

These timings match the decrease in potentially suitable Altai habitat for Denisovans and increase for Neanderthals between ~130 and ~110 ka (Fig. 8f), based on simulations of Denisovan and Neanderthal habitat, in which temperature fluctuations exert a major influence on vegetation[70]. The period of overlap in habitat preferences partly coincides with the modelled age estimate for Denisova 11 (118–79 ka)[2]. The simulations suggest that potentially suitable habitat for both hominins, particularly Denisovans, occurred throughout their period of existence in the Altai, but that habitat was least favourable for Denisovans and most suitable for Neanderthals during the last two interglacial periods (Fig. 8f). The switch from Denisova 2- or 8-like mtDNA sequences before 150 ka to Denisova 3- and 4-like sequences after 80 ka (or 100 ka) may thus reflect the replacement of a Middle Pleistocene population of Denisovans with a Late Pleistocene population. The latest Denisovans are represented by three fossils (Denisova 3, 4 and 22) and mtDNA fragments in sediments deposited up to ~55 ka.

Occupation of the cave by Denisovans and Neanderthals during relatively warm and cold periods and their adaptation to the climatic transitions approximately 200–170 and 130–100 ka, both of which coincided with major turnovers in the mammalian fauna (Fig. 10), may have been aided by the location of Denisova Cave in a hominin refugium[71]. Favourable habitats for modern humans[72] and Denisovans[70] (Fig. 8f) existed across northern Asia during the last glacial period. The disappearance of Denisovans from the Altai without an accompanying faunal turnover may, therefore, be related primarily to demographic factors or to competition for resources or encounters with modern humans, rather than environmental pressures.

## Methods

### Permits and permissions

Excavations were conducted and samples collected during the 2012, 2016, 2017, 2018 and 2019 field seasons. Permits to conduct archaeological excavations at Denisova Cave were issued to M.V.S. by the Ministry of Culture of the Russian Federation for excavations as follows: permit number 210 (issued 15 May 2012), permit number 646 (issued 31 May 2016), permit number 538 (issued 31 May 2017), permit number 1193 (issued 11 May 2018) and permit number 0432-2019 (issued 6 May 2019). Export permits for the sediment samples were obtained from the Novosibirsk Customs Office, Russian Federation, with release dates of 27 August 2013 (for the 2012 samples), 18 April 2017 (for the 2016 samples), 16 April 2018 (for the 2017 samples) and 27

July 2021 (for the 2018 and 2019 samples); no other document particulars are provided. Import permits to Australia were issued to R.G.R. by the Australian Government Department of Agriculture, Fisheries and Forestry (permit number IP12003422 issued 1 March 2012, for the 2012 samples), Department of Agriculture and Water Resources (permit number 0000480685 issued 21 June 2016, for the 2016 and 2017 samples) and Department of Agriculture, Water and the Environment (permit number 0004378957 issued 10 July 2020, for the 2018 and 2019 samples).

An agreement of scientific cooperation between the Institute of Archaeology and Ethnography, Siberian Branch of the Russian Academy of Sciences and the University of Wollongong for projects in the field of geochronology was first signed on 28 March 2012, with the most recent variation signed on 29 January 2019. An agreement of scientific cooperation between the Institute of Archaeology and Ethnography, Siberian Branch of the Russian Academy of Sciences and the Max Planck Institute for Evolutionary Anthropology for projects in the field of palaeogenetics in North Asia was signed on 25 December 2018, with the most recent variation signed on 18 September 2023.

## Collection of sediment samples

Sediment samples for optical dating were collected by hand at night using a red-light torch for illumination, and sealed in thick black plastic to prevent light exposure during transport to the University of Wollongong[1]. Samples for sediment DNA analysis were collected from Profiles A and B[3] and the exposed deposits in Profiles C and D were sampled in a similar, grid-like pattern, deviating only to avoid large rocks and layer boundaries. We used the same sampling and recording procedures as described in ref. 3, including precautions to minimize contamination by modern DNA (sterile gloves, protective face masks, hair nets) and potential cross-contamination of samples (sterilized scalpel blades and zip-lock plastic bags for individual samples)[3]. The same precautions were taken when collecting sediment DNA samples from directly beneath the mammoth ivory figurine[40], which was washed immediately after excavation. The freshly excavated deer tooth pendant was sealed in a plastic bag and the attached clumps of sediment were manually removed in the laboratory using a flexible, disposable plastic microspatula[31].

## Optical dating of sediments

Optical dating gives an estimate of the time since mineral grains of quartz and K-feldspar were last exposed to sunlight[73–75]. The equivalent dose ($D_e$, the radiation energy absorbed by grains since deposition) is divided by the environmental dose rate (the rate of supply of ionizing radiation to the grains since deposition) to determine the optical age (the time of sediment deposition). We used the same or similar sample preparation, measurement and data-analysis procedures as described in ref. 1 (details in Supplementary Section 2).

$D_e$ values were estimated for individual grains of quartz and K-feldspar using the optically stimulated luminescence (OSL) and post-infrared infrared stimulated luminescence (pIRIR) signals, respectively. Single-grain measurements enable grains with unsuitable OSL and pIRIR characteristics to be identified and removed before age determination and for the potential impact of processes such as insufficient exposure of grains to sunlight prior to deposition (partial bleaching) and mixing of sediments after deposition to be investigated[1,60,74]. Both minerals were measured using a single-aliquot regenerative-dose procedure, with $D_e$ and least-squares normalized $L_n/T_n$ values (a precursor to $D_e$ estimation) determined for many samples using standardized growth curves and the $L_nT_n$ method[76–79]. The OSL signal was saturated in samples older than 100–150 ka, so samples older than this were dated using only K-feldspar. For five of the oldest samples, $D_e$ values were estimated using a multiple-aliquot regenerative-dose procedure for K-feldspar[80]; statistically consistent

$D_e$ values were obtained for two of these samples using the single-grain $L_nT_n$ method.

For samples with single-component distributions of $D_e$ or least-squares normalized $L_n/T_n$ values, the final $D_e$ values and uncertainties (standard error of the mean) were estimated using the central age model, after rejecting any outliers identified using the normalized median absolute deviation[81,82]. For $D_e$ and least-squares normalized $L_n/T_n$ distributions consisting of multiple, discrete components, we used the finite mixture model to estimate the mean and standard error of each component and the relative proportion of grains in each of them[82,83]. The minimum age model[81,82] was used in place of the finite mixture model for four samples to estimate the mean and standard error of the population of grains with the smallest $D_e$ or least-squares normalized $L_n/T_n$ values. We consider the most reliable estimates of the time of deposition of each layer to be those obtained for samples that have single-component distributions, or multi-component distributions in which one of the components contains 70% or more of the grains. This threshold value was chosen to include samples composed mostly of grains deposited at around the same time, but it has a negligible effect on the final chronology.

Quartz and K-feldspar grains were measured for 35 samples as a test of internal consistency, and reliable ages were obtained from both minerals for 28 of these samples (Supplementary Fig. 21a). Each of the paired OSL and pIRIR ages is consistent at the 95% CI, and the mean ratio for all 28 samples is statistically consistent with unity ($1.016 \pm 0.015$, standard error of the mean). This implies that the sediments were well bleached by sunlight at the time of deposition, as the pIRIR signal is less light-sensitive than the OSL signal[74,84,85]. For each of these samples, we combined the paired ages to determine the weighted mean age. To do so, we determined the shared and unshared error components of the total age uncertainties, then calculated the weighted mean of each pair of ages (weighted by the inverse square of the unshared errors) and, finally, added in quadrature the average relative shared error ($4.00 \pm 0.03\%$) to estimate the total uncertainty on the weighted mean age.

The total environmental dose rates were estimated from field and laboratory measurements of the external gamma and beta dose rates, respectively, in addition to the contributions from cosmic rays and radioactive emitters internal to the quartz and K-feldspar grains. To estimate the latter, we analyzed 116 individual mineral grains from two samples using quantitative evaluation of mineral–energy dispersive spectroscopy (QEM-EDS). The whole-of-grain arithmetic mean K concentration determined from 115 of these grains ($12.4 \pm 0.9$ wt%) was used to calculate the internal dose rates for all K-feldspar grains in this study (Supplementary Fig. 20d). The final $D_e$ value for each sample was divided by the corresponding total dose rate to estimate the optical age in calendar years before present, with an associated uncertainty that includes all known and estimated sources of random and systematic error.

We developed two Bayesian models of the depositional chronology in South Chamber (details in Supplementary Section 2.5). The measured (unmodelled) and modelled ages and uncertainties are listed in Supplementary Data 14 (model A) and Supplementary Data 15 (model B). Model A included the 31 ages considered the most reliable estimates of the time of deposition of the relevant layer (Fig. 5; Supplementary Figs. 23 and 25d; Supplementary Code 1). Model B included 19 additional ages (i.e., a total of 50 ages) to further constrain the timing of the Middle and Upper Palaeolithic archaeological phases (Supplementary Figs. 24 and 25e; Supplementary Code 2). Modelling was performed on the OxCal platform (version 4.4.4) using the sequence of stratigraphic layers (model A) or archaeological phases (model B) as prior information[86,87]. We modelled the sequence as a series of phases, allowing for the existence of time gaps due to erosional events, periods of little or no sediment deposition, or the absence of samples.

## Ancient sediment DNA data generation

The 235 newly collected sediment samples were transported to the Max Planck Institute for Evolutionary Anthropology in Leipzig, Germany for genetic analysis. Subsamples of between 20.1 and 192.7 mg of sediment were taken from each sample. DNA extraction was performed following the protocol outlined in ref. 88. For subsamples <100 mg, 1 mL of extraction buffer was used and 2 mL was used for larger subsamples (Supplementary Data 16). Of the resulting lysates, 150 μl were used for subsequent DNA purification using binder buffer 'B'[88] and the complete volume of extract (30 μl) was used for single-stranded DNA library preparation[89]. Hybridization captures were performed for each library using mammalian mtDNA and human mtDNA probes[90–92]. DNA purification, library preparation and hybridization enrichment were performed using a Bravo NGS workstation B. Aliquots of 5 μL from each enriched library were pooled and sequenced on an Illumina MiSeq v3 platform using 76 cycle paired-end reads. The software Bustard (Illumina) was used for base calling.

## Identification of ancient mammalian taxa and hominin DNA fragments

The analysis of the resulting capture data was performed using a previously described data analysis pipeline[17]. In brief, LeeHom[93] was used to merge overlapping paired-end reads, which were then mapped to either 242 mammalian mtDNA genomes[91] or the revised Cambridge Reference sequence (human mtDNA)[94] using bwa-aln[95] with parameters adjusted for ancient DNA sequences ('-n 0.01 -o 2 -l 16,500'). Filtering was performed by removing unmapped reads, sequences shorter than 35 base pairs and PCR duplicates. For mammalian mtDNA capture data, PCR duplicates were removed by collapsing identical sequences to a single representative sequence and, for further analysis, we retained only sequences seen at least twice. For human mtDNA capture data, bam-rmdup (https://github.com/mpieva/biohazard-tools) was used to remove duplicates based on start and end alignment coordinates. BLAST[96] and MEGAN[97] were then used to assign the resulting unique reads to mammalian families. The sequences assigned to each family were mapped to all available mtDNA reference genomes for that family, and the reference genome with the highest number of aligned sequences was retained for further analysis. Last, sequences with a mapping quality below 25 were filtered out and residual duplicate sequences removed with bam-rmdup.

To confirm the presence of ancient DNA sequences from a given biological family the following criteria were applied: (1) at least 1% of the total number of fragments assigned to mammalian families had to be assigned to the given family; (2) at least 10 of the assigned fragments had to be putatively deaminated (carried C-to-T substitutions at the 5' or 3' termini); and (3) the observed terminal C-to-T substitution frequencies had to be significantly higher than 10% based on 95% binomial confidence intervals calculated with R version 3.5.1.

Sub-family assignments on the level of mitochondrial groups were performed for hominin, ursid and hyaenid sequences using previously described diagnostic positions among published mtDNA genomes[3]. Hominin group assignments (Neanderthal, Denisovan or ancient modern human), ursid assignments (brown, black, polar, cave, moon, panda, short-faced, sloth, spectacled, or sun bear) and hyaenid assignments (cave and spotted (haplogroup A), cave haplogroup B, spotted (haplogroup C), cave haplogroup D, striped, or brown hyaena) were considered positive if they were supported by significantly more than 10% of the sequences based on 95% binomial confidence intervals and by sequences sharing the derived state of the group at three or more unique diagnostic positions. Group assignments were performed using all sequences, except for the identification of modern human sequences, where only deaminated sequences were considered in order to exclude present-day human contamination.

Sequence assignments to specific Neanderthal and Denisovan mtDNA lineages were performed using kallisto[67] as previously described[21] for samples with at least 250 ancient hominin sequences. The total number of endogenous hominin sequences was calculated by subtracting the estimated number of contaminant (modern human) sequences from the total number of unique hominin sequences. The number of contaminant sequences was calculated as the number of sequences supporting diagnostic positions specific to modern humans in the mtDNA (Supplementary Data 18). Assignment of at least 20% of the identified sequences to a lineage was required for a lineage identification. For each sample, the number of mtDNA sequences identified as similar to Denisova 3 or Denisova 4 were added together, as were those identified as similar to Denisova 2 or 8.

## Reporting summary

Further information on research design is available in the Nature Portfolio Reporting Summary linked to this article.

## Data availability

All data for optical dating are provided in Supplementary Figs. and Supplementary Data. Previously published data are provided in Supplementary Figs. and Supplementary Tables in ref. 1. All optical dating data and samples are stored in the Optical Dating Facility at the University of Wollongong. Any other relevant data are available from Z.J. and B.L. upon reasonable request. All sequence data from the mammalian and human mtDNA captures are available in the European Nucleotide Archive under accession number PRJEB80323. Previously published data are provided in ref. 3 and are available as follows: mtDNA consensus sequences reported from Main Chamber layers 19 (M65) and 20 (M71), and from East Chamber layers 11.4 (E202) and 11.4/ 12.1 (E213), are available in the Dryad digital repository (https://doi.org/ 10.5061/dryad.k3j9kd567), and the raw data for each mammalian mtDNA and human mtDNA enriched library are available in the European Nucleotide Archive under accession number PRJEB44036. All sediment DNA data and samples are housed in the Max Planck Institute for Evolutionary Anthropology. Any other relevant data are available from E.Z. and M.M. upon reasonable request.

## Code availability

Luminescence data analyses were performed using functions implemented in R packages *numOSL* and *Luminescence* (details supplied in Supplementary Section 2.3), the CQL codes used for Bayesian modelling of the optical ages are provided in Supplementary Code 1 and Supplementary Code 2, and the mtDNA data were processed using packages available at https://github.com/mpieva.

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

## Acknowledgements

This project was funded by the Australian Research Council (fellowships FT150100138 to Z.J., FT140100384 to B.L. and FL130100116 to R.G.R.), the European Research Council (grant agreement number 694707 to S.Pääbo), the Russian Science Foundation (grant number 24-18-00069 to M.V.S., M.B.K., V.A.U., A.K.A. and S.K.V.) and the Max Planck Society. E.I.Z. was partially supported by the Miller Institute for Basic Research in Science (University of California, Berkeley), and K.O. was supported by an Australian Government Research Training Programme Award. Genetic data were produced by the Ancient DNA Core Unit of the Max Planck Institute for Evolutionary Anthropology, which is funded by the Max Planck Society. Microscopy Australia at the Centre for Advanced Microscopy, Australian National University, provided access to QEM-EDS facilities and scientific and technical assistance. We thank Y. Jafari and T. Lachlan for laboratory assistance with the optical dating, V. Vaneev for fieldwork support, J. Lihanova, F. Müller and E. Essel for support with sampling for sediment DNA analysis, F. Romagné for assistance with DNA data processing, W. Saktura for creating the globe image in Fig. 1, and J. Ruan for providing the habitat suitability data and related information.

## Author contributions

Z.J., E.I.Z., S.Pääbo, M.M. and R.G.R. designed this study. M.V.S., M.B.K. and A.P.D. led the excavations and artefact analyses. Z.J., B.L., K.O. and R.G.R. collected the optical dating and sediment DNA samples, and P.G. collected the micromorphology samples. M.V.S., M.B.K., A.P.D., V.A.U. and P.G. provided archaeological and geological context and interpretation. Z.J., B.L. and K.O. conducted the optical dating, with input from R.G.R. K.O. and F.B. obtained the QEM-EDS data. E.I.Z., S.Peyrégne, V.S., J.K. and M.M. performed, aided in or supervised analysis and visualization of the DNA data. Other analyses were led by V.A.U. (stratigraphy and sedimentology), P.G. (micromorphology), A.K.A. (small vertebrates) and S.K.V. (large mammals). Z.J., E.I.Z., J.K., M.M. and R.G.R. wrote the manuscript with contributions from all authors.

## Funding

## Competing interests

The authors declare no competing interests.
