## [Transparent Peer Review file · Nature Communications]

Pleistocene chronology and history of hominins and fauna at Denisova Cave

Corresponding Author: Dr Matthias Meyer

Version 0:

Reviewer comments:

Reviewer #1

(Remarks to the Author)

The is an excellent and welcome expansion of the previous and influential studies that integrate aDNA, geochronology and geoarchaeology at Denisova. Here the authors provide a holistic view of the entire, excavated cave sequence. The results, while providing no great surprises, are inspiring in terms of the quality of the work, in particular the integration of various lines of evidence to produce an exciting demographic story of Neanderthals and Denisovan occupation.

As a geoarchaeologist, I will focus my review generally on the geoarchaeological/micromorphological portion of the manuscript.

The micromorphological data is clearly presented within the text and it seems obvious that it, together with the field descriptions and microstratigraphic interpretations, formed the basis for the interpretation of the chronology and aDNA results. There were a few minor points I would like to make:

The authors note that layers 12 and 9 were „disturbed by burrowing and phosphate mineralization.“ The way this is written it implies that there is physical disturbance from both processes. Is phosphate mineralization (i.e., nodule and vein growth) physically displacing/distorting the groundmass? Does phosphatization play any role in volume reduction of the sediments or is the slumping and deformation purely a result of subsidence through the inferred swallow hole?

The authors state that layer 12 in profile D is not phosphatized. Does that mean that the other layers in profile D ARE phosphatized? This was a bit confusing because from figure 3 it seems that the highest success-rate for hominin aDNA recovery in Profile D was from layer 13. This would be surprising if 13 is extensively phosphatized. Given the significance that the authors afford to degree of phosphatization in later interpretations of aDNA recovery from sediments, it may be helpful to provide clearer statements about which layers (or zones) are extensively phosphatized or not.

I would call them “phosphatized ashes” instead of phosphatic ashes

In figure 2, is there any way that you could include sample numbers for micromorph blocks in the figures? In trying to piece the micromorphology story together, I had to go from figure to figure, and find this information only in the caption. The blocks are small and there are already quite a bit of labeled boxes there, but it would be better to have that information in the image and not the caption.

In extended data figure 2, the scale of scans of thin sections are presumably cms and not mm?

My one major comment would be that the figures for the micromorphology part could be improved a bit. I get that there is limited space (at least within the body of the manuscript and the extended data figures), but the current extended data figure 2 has a lot of images at a range of scales and it is a bit difficult to decipher everything. Furthermore, I don't think that the selected images clearly show what the authors are trying to illustrate. Curiously, the figures (both here and the supplemental figure) seem to rely on scans of thin sections and what appear to be low-magnification photomicrographs. This works well for illustrating larger-scale features (i.e., zones of gleying), but is less helpful for things like the crushed bone, or lenticular/platy microstructures, rounded aggregates, etc. Since there is room in the supplemental, it would be nice if the key features discussed in the text are more fully illustrated with high-quality and higher resolution photomicrographs, including in XPL. Would it be possible to make extended data figure 2 less “busy,” by selecting just a few, key images and then fill out the supplemental with more detailed images of the features mentioned in the text?

Reviewer #2

(Remarks to the Author)

This is a further installment of a series of articles on Denisova Cave, containing Denisovian and Neanderthal archaic humans, in southern Siberia. This one concentrates on the previously unreported South Chamber. It basically follows the procedures developed in the previous articles and as such is mainly an application. As was the case in the previous articles, the work is exhaustive and extensively researched, to the credit of the authors. The paper is extremely data rich, which is important for documentation of this important site. I glanced through the data and could not find any obvious problems. My comments are minor, mainly to provide clarification.

Fig 1. It would be useful to show where the rock spill ridge is in relation to the profiles. I am not sure I see it in Profiles B-D, although presumably they cross it.

Fig 2. The caption says the layer colors are notational, but I do not know what that means – just to distinguish them?

Fig. 5. The diatom record for temperature and the local climate from fauna do not seem to correlate. Any comment?

Extended data Fig. 7. Is the x-axis in percentage or fraction? I assume the latter but the title says percentage.

In Methods, Optical Dating:

When you say the 70% threshold was used to include multi-component samples in the final chronology, but that is has negligible effect on that chronology, do you mean that if you left all those samples out, the conclusions would be the same? Please clarify.

Which pIRIR procedure did you use and are you convinced any fading or recuperation is negligible? Please elaborate.

I assume you did not use QEM-EDS on every feldspar grain. Please explain your sampling strategy. Were they all K-feldspar grains? (I realize this is explained in more detail in the supplement, but some mention should be given in the main text.)

Model B Bayesian analysis included 19 additional samples from those used in Model A. Are these samples that were multi-component but had at least 70% in one component, or what? Also for Model B you used archaeological phases as prior information. Are these phases based on artifacts and how discrete are they? I imagine there would be more overlap than would be the case with stratigraphy. Please justify.

Genetic dating is mentioned in a couple of places in the text. You should include in the Methods section how this was done. Is it based on assumed constant rate of mtDNA change?

Supplement:

Section 1: Could you provide the criteria by which layers were distinguished?

Section 2: Luminescence dating. The first sentence is not strictly true. While luminescence provides dates of sediment burial, it does not date associated materials, depending I suppose on what you mean by “associated”. Dating a sediment layer does not necessarily date what is in the layer, which could be older or younger. Please use more precise language. In the quartz analysis, how did you account for the possibility of heterogeneity in the beta dose rate? (I think it is taken up later, but there should be some mention here.)

For the feldspars, I think you need to flesh out more the three methods (A, B, C) instead of just supplying a reference. Give a short description of each method to clarify how D_e results were obtained. I do not understand, for example, how a multi-aliquot procedure (C) results in a distribution of $\ln(T_n)$ values (Fig. 16). What pIRIR method was used (preheat, stimulation temperature, number of high temperature stimulations, etc.) and was this method sufficient to reduce fading and recuperation to negligible levels?

In the discussion of MAM, you say for three quartz samples the D_e was determined by MAM, but then say it was determined by FMM. This is confusing.

You say at one point that beta and gamma dose rates are measured by emission methods (thus low error if disequilibrium), but then say U and Th was measured by ICP-MS, which detects parent atoms. Then you talk about possible disequilibrium. Try to make your language more consistent. Also I wonder if U-238 and U-234 fractionization, or even radium migration, might be more likely to cause the discrepancies than radon loss. Either of these might have more effect on the dose rate, depending on when they occurred, even when using beta counting for the beta dose rate.

You mention that spatial variation in beta emitters may account for some of the distribution of D_e values, but do you take that into account when determining the D_e used for age calculation?

Jim Feathers
Univ. of Washington

Reviewer #3

(Remarks to the Author)

In this study, Jacobs and colleagues present the results of new excavations in the South Chamber of Denisova Cave. The authors provide a detailed record of the cave stratigraphy, generate 49 new optical ages, and recover mitochondrial DNA from 235 sediment samples. Combined with previously published data from other chambers, this impressive dataset offers a comprehensive overview of the cave's occupancy and climate history over the last 300,000 years.

The manuscript is well-written and presents valuable new insights into this enigmatic cave. Overall, I find this work highly interesting and believe it will be of significance to the broader scientific community.

Below, I have outlined a few minor suggestions and points for clarification. I should note, however, that I am not an expert in

optical dating and therefore cannot comment on the specific methods used for this data.

- The current manuscript lacks a “data availability” statement. Access to the mitochondrial capture data and to the pipelines and scripts would greatly benefit the wider scientific community.
- The figures are detailed and effectively represent the data and results. However, some could benefit from additional labeling. For instance, in Figures 4 and 5, including small in-figure legends clarifying the colors would aid in faster interpretation.
- Perhaps I overlooked something, but in Figure 5, the inferred climate patterns do not always overlap between the different chambers. Is there an explanation for this discrepancy?
- The authors perform sub-family assignments for hominin, ursid, and hyaenid sequences, even though some layers are dominated by other families, such as bovids, which I assume were common in hominin diets. While not essential for the robustness of the study, exploring changes in these other lineages could provide additional insights.
- The supplementary sample summary seems to contain some formatting errors in the rows for positive and negative controls.

Version 1:

Reviewer comments:

Reviewer #1

(Remarks to the Author)

The authors have addressed all concerns and suggested revisions, and the current manuscript is appropriate for publication.

Reviewer #2

(Remarks to the Author)

This is a very thoroughly researched paper. In my previous review my comments were mainly for clarification. I read through the revision carefully and am satisfied by concerns have been addressed.

Reviewer #3

(Remarks to the Author)

The authors have addressed all my questions.

We are grateful to the reviewers for their encouraging comments and thoughtful suggestions for possible improvements. Our responses to each of their comments are shown below in red text. In the revised ms for the main paper and in Supplementary Information, we have shown alterations to the text using 'track changes' (edits are coloured red). The revised ms includes five additional figures in the main paper and a further nine figures in Supplementary Information. So, for clarity, we have not highlighted the altered figure numbers using track changes, and the same is true for the revised reference numbers in Supplementary Information.

Reviewer #1 (Remarks to the Author):

The is an excellent and welcome expansion of the previous and influential studies that integrate aDNA, geochronology and geoarchaeology at Denisova. Here the authors provide a holistic view of the entire, excavated cave sequence. The results, while providing no great surprises, are inspiring in terms of the quality of the work, in particular the integration of various lines of evidence to produce an exciting demographic story of Neanderthals and Denisovan occupation.

As a geoarchaeologist, I will focus my review generally on the geoarchaeological/micromorphological portion of the manuscript.

The micromorphological data is clearly presented within the text and it seems obvious that it, together with the field descriptions and microstratigraphic interpretations, formed the basis for the interpretation of the chronology and aDNA results. There were a few minor points I would like to make:

The authors note that layers 12 and 9 were „disturbed by burrowing and phosphate mineralization.” The way this is written it implies that there is physical disturbance from both processes. Is phosphate mineralization (i.e., nodule and vein growth) physically displacing/distorting the groundmass? Does phosphatization play any role in volume reduction of the sediments or is the slumping and deformation purely a result of subsidence through the inferred swallow hole?

Burrowing has certainly been an important process of physical disturbance of layers 9 and 12. There have also been multiple phases of phosphatization of these uppermost Pleistocene layers. Localized occurrences of phosphatic (apatite) crusts/rimms around limestone grains in sample DEN18-18 (Fig. 3f), for example, are similar to features observed in parts of the older deposits in Main and East Chambers (ref. 27), so they seem to have formed throughout the cave history. Holocene-derived phosphates are responsible for the large nodules formed in pdd-9 and -12, and have resulted in localized displacements of the groundmass. The deeper Pleistocene sediments are most affected by subsidence (as described in the reconstructed sequence of sediment deposition and deformation in Supplementary Section 1.1), with signs of slickensides within the phosphatized groundmass indicating that subsidence took place after phosphatization.

The authors state that layer 12 in profile D is not phosphatized. Does that mean that the other layers in profile D ARE phosphatized? This was a bit confusing because from figure 3 it seems that the highest success-rate for hominin aDNA recovery in Profile D was from layer 13. This would be surprising if 13 is extensively phosphatized. Given the significance that the authors afford to degree of phosphatization in later interpretations of aDNA recovery from sediments, it may be helpful to provide clearer statements about which layers (or zones) are extensively phosphatized or not.

Some parts of layer 12 have been extensively phosphatized and we have referred to these parts as pdd-12. Other parts of layer 12 as well as the underlying layers, including layer 13, have not been noticeably affected by phosphatization. We have attempted to clarify this distinction in the revised text to the section titled 'Stratigraphy and micromorphology'.

I would call them “phosphatized ashes” instead of phosphatic ashes
Done (and likewise with 'phosphatic rimms').

In figure 2, is there any way that you could include sample numbers for micromorph blocks in the figures? In trying to piece the micromorphology story together, I had to go from figure to figure, and find this information only in the caption. The blocks are small and there are already quite a bit of labeled boxes there, but it would be better to have that information in the image and not the caption.

Done. We have retained some of the original information in the figure caption, for clarity.

In extended data figure 2, the scale of scans of thin sections are presumably cms and not mm?

Indeed. Thank you for spotting this oversight, which we have now corrected and cross-checked in all of the micromorphology figures (see below).

My one major comment would be that the figures for the micromorphology part could be improved a bit. I get that there is limited space (at least within the body of the manuscript and the extended data figures), but the current extended data figure 2 has a lot of images at a range of scales and it is a bit difficult to decipher everything. Furthermore, I don't think that the selected images clearly show what the authors are trying to illustrate. Curiously, the figures (both here and the supplemental figure) seem to rely on scans of thin sections and what appear to be low-magnification photomicrographs. This works well for illustrating larger-scale features (i.e., zones of gleying), but is less helpful for things like the crushed bone, or lenticular/platy microstructures, rounded aggregates, etc. Since there is room in the supplemental, it would be nice if the key features discussed in the text are more fully illustrated with high-quality and higher resolution photomicrographs, including in XPL. Would it be possible to make extended data figure 2 less "busy," by selecting just a few, key images and then fill out the supplemental with more detailed images of the features mentioned in the text?

We have now increased substantially the number and type of micromorphology images in the revised ms, following the helpful suggestions of this reviewer. There is a new composite figure in the main paper (Fig. 3) that shows a thin-section scan, PPL photomicrograph and XPL photomicrograph for each of two samples: one from the deep deposits in Profile A (DEN18-4) and one from the uppermost Pleistocene deposits between Profiles B and C (DEN18-18). In Supplementary Section 1.1, we have also included three new micromorphology figures (Supplementary Figs. 1–3) that contain a variety of thin-section scans and PPL and XPL photomicrographs of the other South Chamber samples. Furthermore, we have included two additional photomicrographs of the East Chamber sample in Supplementary Fig. 4 (originally Supplementary Fig. 1) to display some of the details of the key features of interest. We think that the resolution of the images in these figures is sufficient to clearly observe the features described in the text and captions.

Reviewer #2 (Remarks to the Author):

This is a further installment of a series of articles on Denisova Cave, containing Denisovian and Neanderthal archaic humans, in southern Siberia. This one concentrates on the previously unreported South Chamber. It basically follows the procedures developed in the previous articles and as such is mainly an application. As was the case in the previous articles, the work is exhaustive and extensively researched, to the credit of the authors. The paper is extremely data rich, which is important for documentation of this important site. I glanced through the data and could not find any obvious problems. My comments are minor, mainly to provide clarification.

Fig 1. It would be useful to show where the rock spall ridge is in relation to the profiles. I am not sure I see it in Profiles B-D, although presumably they cross it.

Done. We have now included an area of grey shading in Fig. 1 to indicate the approximate location of the ridge of spalled rocks in South Chamber.

Fig 2. The caption says the layer colors are notational, but I do not know what that means – just to distinguish them?

This is correct. To avoid confusion, we have now replaced the word 'notional' with 'arbitrary' in the figure caption, so that it now reads "Stratigraphic layers are distinguished by arbitrary colours".

Fig. 5. The diatom record for temperature and the local climate from fauna do not seem to correlate. Any comment?

The Lake Baikal and Denisova Cave climatic records likely differ due to differences in scale (regional versus local, respectively) and to factors that affect interpretation of the cave records. We have expanded on

these caveats in two new paragraphs at the end of the section titled 'Sediment DNA of ancient mammals', incorporating some of the text originally in the 'Discussion'.

Extended data Fig. 7. Is the x-axis in percentage or fraction? I assume the latter but the title says percentage.

Thank you for spotting this oversight. The numbers on the x-axis should, indeed, have been shown as percentages, which we have now done in the revised figure (now Supplementary Fig. 29).

In Methods, Optical Dating

When you say the 70% threshold was used to include multi-component samples in the final chronology, but that it has negligible effect on that chronology, do you mean that if you left all those samples out, the conclusions would be the same? Please clarify.

Yes, the results are indistinguishable, but the precision is better for model A because more data are included. We ran the model with those samples included (model A: Fig. 5 and Supplementary Fig. 23) and with them excluded (model B: Supplementary Fig. 24). The results for both models are also provided in Supplementary Tables 14 and 15 (models A and B, respectively). The differences between the two models are fully explained in Supplementary Section 2.5, which we cite in Methods.

Which pIRIR procedure did you use and are you convinced any fading or recuperation is negligible? Please elaborate.

We now provide this information, along with additional references, in Supplementary Section 2.3 ('D_e estimation for K-feldspar). Experimental results for both fading and recuperation tests on samples from Denisova Cave have been published previously (ref. 1). In that paper, we explain why corrections to the pIRIR data are thought not to be necessary. We have now added a new paragraph to Supplementary Section 2.3 indicating that residual dose and fading tests were conducted in our earlier study, and that we applied no corrections to any of the South Chamber samples in the present study.

I assume you did not use QEM-EDS on every feldspar grain. Please explain your sampling strategy. Were they all K-feldspar grains? (I realize this is explained in more detail in the supplement, but some mention should be given in the main text.)

This is correct. We applied QEM-EDS to two samples, one from the Upper Palaeolithic levels (DCS17-6) and the other from the Middle Palaeolithic levels (DCS17-3). Details are provided in Supplementary Section 2.4 ('Internal dose rates'). We have changed the wording in Methods to specify the number of samples and grains that were analyzed using QEM-EDS and to cite the relevant figure in Supplementary Information (Supplementary Fig. 20d). As regards mineral composition, Supplementary Table 11 gives a breakdown of the different mineral phases identified for the 116 grains analyzed using QEM-EDS: 97.7% and 98.6% area proportions of K-feldspar within grains for samples DCS17-6 and DCS17-3, respectively.

Model B Bayesian analysis included 19 additional samples from those used in Model A. Are these samples that were multi-component but had at least 70% in one component, or what?

Not necessarily. The difference between the two models is the inclusion of 19 samples that have a clear association with a specific archaeological phase, but not with a specific stratigraphic layer (e.g., samples from the dMP deposits associated with the middle Middle Palaeolithic). As we note in Supplementary Section 2.5 and in the caption for Supplementary Fig. 24, these 19 samples have D_e distributions dominated by a single population of grains or by a component that contains more than 70% of the grains. We have slightly expanded the text in Supplementary Section 2.5 to clarify the distinction between models A and B.

Also for Model B you used archaeological phases as prior information. Are these phases based on artifacts and how discrete are they? I imagine there would be more overlap than would be the case with stratigraphy. Please justify.

There is an extensive literature on the archaeological phases represented at Denisova Cave (e.g., refs. 4–7 and 36–52). The number of artefacts recovered from the early Middle Palaeolithic, middle Middle Palaeolithic and Upper Palaeolithic layers in South Chamber between 2017 and 2022 are listed in

Supplementary Table 2, and a discussion of the types of artefacts recovered from the cave deposits is given in Supplementary Section 1.3 and illustrated by Supplementary Figs. 5–10. Additional details about the different archaeological phases are available in our previous paper (ref. 1). To alert readers to the relevant section of the present paper, we now refer to Supplementary Section 1 in the opening section of the main text, where we first mention the archaeological assemblages in South Chamber.

Genetic dating is mentioned in a couple of places in the text. You should include in the Methods section how this was done. Is it based on assumed constant rate of mtDNA change?

We present no new genetic dates in this ms, and all references to molecular dates are from previous publications, so we have not included a description in Methods. We instead cite the earlier studies that estimated genetic dates obtained for the deer tooth pendant and Denisova 4; these studies include full descriptions of the genetic dating methods. In short, the genetic date for the pendant (layer 11) was estimated using BEAST2 and applying a relaxed log normal mutation clock that allows for mutation rate variation across the phylogeny. The genetic date estimate for Denisova 4 is the range estimated by a Bayesian age model that incorporates multiple sources of information, including both archaeological and genetic data.

Supplement:

Section 1: Could you provide the criteria by which layers were distinguished?

Done. We have now expanded the first paragraph of Supplementary Section 1.1 ('Stratigraphy and sedimentology of Pleistocene deposits in South Chamber') to include descriptions of the excavation methods and the criteria used to distinguish the individual layers; these are same as those used in our earlier study (ref. 1). Summary descriptions of each of the layers in South Chamber are also provided in Supplementary Table 1.

Section 2: Luminescence dating. The first sentence is not strictly true. While luminescence provides dates of sediment burial, it does not date associated materials, depending I suppose on what you mean by "associated". Dating a sediment layer does not necessarily date what is in the layer, which could be older or younger. Please use more precise language.

Done. We have now reworded the first sentence to emphasize that the association only holds true if the artefacts, fossils and other incorporated materials are in primary depositional context. The amended sentence at the start of Supplementary Section 2 now reads as follows: "Optical dating provides a means of determining burial ages for sediments and associated artefacts, fossils and other materials deposited at the same time as the sediments and not translocated subsequently^{1,30–33}."

In the quartz analysis, how did you account for the possibility of heterogeneity in the beta dose rate? (I think it is taken up later, but there should be some mention here.)

We did not make any formal correction for beta dose heterogeneity, but we note at the end of the subsection titled 'External beta dose rates' (Supplementary Section 2.4) that our Timepix experiments on two samples from Denisova Cave (ref. 62 in Supplementary Information) showed that the extent of scatter in D_e values and the shape of the D_e distributions can be fully or partly explained by beta microdosimetry variations and other known sources of variability in single-grain D_e values.

For the feldspars, I think you need to flesh out more the three methods (A, B, C) instead of just supplying a reference. Give a short description of each method to clarify how D_e results were obtained. I do not understand, for example, how a multi-aliquot procedure (C) results in a distribution of \ln/T_n values (Fig. 16).

Done. We now provide a short description of each of the three methods in Supplementary Section 2.3 (' D_e estimation for K-feldspar'), in addition to citing method-specific references as well as ref. 1 for full details.

What pIRIR method was used (preheat, stimulation temperature, number of high temperature stimulations, etc.) and was this method sufficient to reduce fading and recuperation to negligible levels?

As per our response above, we now provide this information, along with additional references, in Supplementary Section 2.3 ('D_e estimation for K-feldspar).

In the discussion of MAM, you say for three quartz samples the D_e was determined by MAM, but then say it was determined by FMM. This is confusing.

This was, indeed, very confusing and also incorrect. The MAM was applied to these three quartz samples, not the FMM. We have now deleted the offending sentence near the end of Supplementary Section 2.3.

You say at one point that beta and gamma dose rates are measured by emission methods (thus low error if disequilibrium), but then say U and Th was measured by ICP-MS, which detects parent atoms. Then you talk about possible disequilibrium. Try to make your language more consistent.

We used emission-counting methods for estimation of both beta and gamma dose rates, as described in Supplementary Section 2.4 ('Environmental dose rate determination'). We also compared these beta dose rates (measured using a GM-25-5 beta counter) with those derived from ICP-MS/OES measurements of U, Th and K, and observed a systematic difference. GM-25-5 beta counting does not distinguish between emissions from U, Th and K, so we used the ICP-MS/OES data to investigate some possible reasons for the differences in beta dose rate. The accompanying discussion starts from the paragraph beginning "To further investigate the differences between..." in the subsection titled 'External beta dose rates'. We do not consider our use of language in the discussion to be inconsistent, as we clearly stipulate which measurements were used for which tests and which data sets were used for final age determination.

Also I wonder if U-238 and U-234 fractionization, or even radium migration, might be more likely to cause the discrepancies than radon loss. Either of these might have more effect on the dose rate, depending on when they occurred, even when using beta counting for the beta dose rate.

We agree that there are several possible causes of the systematic difference between the beta dose rates determined from the GM-25-5 and ICP-MS/OES data sets. Many of these, however, cannot be tested using the available data. Understanding and modelling the fractionation of U-238 and U-234, and radium migration, would require data obtained using methods such as high-resolution gamma spectrometry and alpha spectrometry, which we have not applied to the Denisova Cave samples. We clearly state (in the opening paragraph of Supplementary Section 2.4) our assumption that the present-day radionuclide activities and dose rates have prevailed throughout the period of sample burial, and note that modelling of time-dependent disequilibria in the U and Th decay chains in fluvial and cave sediments (refs. 56 and 57 in Supplementary Information) suggests that our approach is unlikely to give rise to errors in the total dose rate of more than a few percent.

You mention that spatial variation in beta emitters may account for some of the distribution of D_e values, but do you take that into account when determining the D_e used for age calculation?

As per our response above, we did not make a correction for beta dose heterogeneity, given the results of our Timepix experiments on two Denisova Cave samples.

Jim Feathers
Univ. of Washington

Reviewer #3 (Remarks to the Author):

In this study, Jacobs and colleagues present the results of new excavations in the South Chamber of Denisova Cave. The authors provide a detailed record of the cave stratigraphy, generate 49 new optical ages, and recover mitochondrial DNA from 235 sediment samples. Combined with previously published data from other chambers, this impressive dataset offers a comprehensive overview of the cave's occupancy and climate history over the last 300,000 years.

The manuscript is well-written and presents valuable new insights into this enigmatic cave. Overall, I find this work highly interesting and believe it will be of significance to the broader scientific community.

Below, I have outlined a few minor suggestions and points for clarification. I should note, however, that I am not an expert in optical dating and therefore cannot comment on the specific methods used for this data.

- The current manuscript lacks a “data availability” statement. Access to the mitochondrial capture data and to the pipelines and scripts would greatly benefit the wider scientific community.

A ‘Data availability’ statement is now included in the manuscript, which indicates the data have been uploaded to the European Nucleotide Archive (accession number PRJEB80323). We have also provided a ‘Code availability’ statement for the mtDNA and geochronology data analyses. The pipeline used for processing the mtDNA data has previously been published (ref. 17).

- The figures are detailed and effectively represent the data and results. However, some could benefit from additional labeling. For instance, in Figures 4 and 5, including small in-figure legends clarifying the colors would aid in faster interpretation.

Done. We have now incorporated small in-figure legends into both Figs. 8 and 10 (originally Figs. 4 and 5, respectively).

- Perhaps I overlooked something, but in Figure 5, the inferred climate patterns do not always overlap between the different chambers. Is there an explanation for this discrepancy?

There are several possible reasons for differences among the three chambers, which we elaborate on in the penultimate paragraph of the section titled ‘Sediment DNA of ancient mammals’. Despite these caveats, however, the overall consistency of the archaeological, hominin, faunal and climatic records across all three chambers is striking.

- The authors perform sub-family assignments for hominin, ursid, and hyaenid sequences, even though some layers are dominated by other families, such as bovids, which I assume were common in hominin diets. While not essential for the robustness of the study, exploring changes in these other lineages could provide additional insights.

We agree that it would be interesting to explore changes in all identified families. We selected hominins, ursids and hyaenids as well-established mtDNA phylogenies are available for these taxa. Performing a similar analysis for families that have less well-established or more complex phylogenies (e.g., bovids) would require substantial additional analyses, which lie beyond the scope of the present study. However, all the data from our study are available for researchers wishing to explore other taxa in greater depth.

- The supplementary sample summary seems to contain some formatting errors in the rows for positive and negative controls.

Thank you for alerting us to these formatting errors. We have now removed the coloured formatting from these rows.